# Parallelizing Thompson Sampling

**Amin Karbasi**
Yale University
amin.karbasi@yale.edu

**Vahab Mirrokni**
Google Research
mirrokni@google.com

**Mohammad Shadravan**
Yale University
mohammad.shadravan@yale.edu

## Abstract

How can we make use of information parallelism in online decision making problems while efficiently balancing the exploration-exploitation trade-off? In this paper, we introduce a batch Thompson Sampling framework for two canonical online decision making problems, namely, stochastic multi-arm bandit and linear contextual bandit with finitely many arms. Over a time horizon $T$, our *batch* Thompson Sampling policy achieves the same (asymptotic) regret bound of a fully sequential one while carrying out only $O(\log T)$ batch queries. To achieve this exponential reduction, i.e., reducing the number of interactions from $T$ to $O(\log T)$, our batch policy dynamically determines the duration of each batch in order to balance the exploration-exploitation trade-off. We also demonstrate experimentally that dynamic batch allocation dramatically outperforms natural baselines such as static batch allocations.

## 1 Introduction

Many problems in machine learning and artificial intelligence are sequential in nature and require making decisions over a long period of time and under uncertainty. Examples include A/B testing [Graepel et al., 2010], hyper-parameter tuning [Kandasamy et al., 2018], adaptive experimental design [Berry and Fristedt, 1985], ad placement [Schwartz et al., 2017], clinical trials [Villar et al., 2015], and recommender systems [Kawale et al., 2015], to name a few. Bandit problems provide a simple yet expressive view of sequential decision making with uncertainty. In such problems, a repeated game between a learner and the environment is played where at each round the learner selects an action, so called an *arm*, and then the environment reveals the reward. The goal of the learner is to maximize the accumulated reward over a *horizon $T$*. The main challenge faced by the learner is that the environment is unknown, and thus the learner has to follow a policy that identifies an efficient trade-off between the exploration (i.e., trying new actions) and exploitation (i.e., choosing among the known actions). A common way to measure the performance of a policy is through *regret*, a game-theoretic notion, which is defined as the difference between the reward accumulated by the policy and that of the best fixed action in hindsight.

We say that a policy has no regret, if its regret growth-rate as a function of $T$ is sub-linear. There has been a large body of work aiming to develop no-regret policies for a wide range of bandit problems (for a comprehensive overview, see [Lattimore and Szepesvári, 2020, Bubeck and Cesa-Bianchi, 2012, Slivkins, 2019]). However, almost all the existing policies are fully sequential in nature, meaning that once an action is executed the reward is immediately observed by the learner and can be incorporated to make the subsequent decisions. In practice however, it is often more preferable (and sometimes the only way) to explore many actions in parallel, so called a *batch of actions*, in order to gain more information about the environment in a timely fashion. For instance, in clinical trials, a phase of medical treatment is often carried out on a group of individuals and the results are gathered for the entire group at the end of the phase. Based on the collected information, the treatment for the subsequent phases are devised [Perchet et al., 2016]. Similarly, in a marketing campaign, the response to a line of products is not collected in a fully sequential manner, instead, a batch of products are mailed to a subset of costumers and their feedback is gathered collectively

[Schwartz et al., 2017]. Note that developing a no-regret policy is impossible without any information exchange about the carried out actions and obtained rewards. Thus, the main challenge in developing a batch policy is to balance between how many actions to run in parallel (i.e., batch size) versus how frequently to share information (i.e., number of batches). At one end of the spectrum lie the fully sequential no-regret bandit policies where the batch size is 1, and the number of batches is $T$. At the other end of the spectrum lie the fully parallel policies where the batch size is $T$ and all the actions are completely determined a priory without any amount of information exchange (such policies clearly suffer a linear regret).

In this paper, we investigate the sweet spot between the batch size and the corresponding regret in the context of Thompson Sampling (TS). More precisely,

- For the stochastic $N$-armed bandit, we develop Batch Thomson Sampling (B-TS), a batch version of the vanilla Thomson Sampling policy, that achieves the problem-dependent asymptotic optimal regret with $O(N \log T)$ batches. B-TS policy with the same number of batches also achieves the problem independent regret bound of $O(\sqrt{NT \log T})$ with Beta priors, and a slightly improved regret bound of $O(\sqrt{NT \log N})$ with Gaussian priors.
- For the stochastic $N$-armed bandit, we develop Batch Minimax Optimal Thompson Sampling (B-MOTS), a batch Thompson Sampling policy that achieves the optimal minimax problem-independent regret bound of $O(\sqrt{NT})$ with $O(N \log T)$ batches. We also present B-MOTS-J, a variant of B-MOTS, designed for Gaussian rewards, which achieves both minimax and asymptotic optimality with $O(N \log(T))$ batches.
- Finally, for the linear contextual bandit with $N$ arms, we develop Batch Thompson Sampling for Contextual Bandits (B-TS-C) that achieves the problem-independent regret bound of $\tilde{O}(d^{3/2}\sqrt{T})$ with $O(N \log(T))$ batches.

The main idea that allows our batch policy to achieve near-optimal regret guarantees while reducing the number of sequential interactions with the environment from $T$ to $O(\log T)$ is a novel *dynamic* batch mechanism that determines the duration of each batch based on an offline estimation of the regret accumulated during that phase. We also observe empirically that batch Thompson Sampling methods with a fixed batch size, but equal number of batches, incur higher regrets.

## 2   Related Work

In this paper, we mainly focus on Thompson Sampling (also known as posterior sampling and probability matching), the earliest principled way for managing the exploration-exploitation trade-off in sequential decision making problems [Thompson, 1933, Russo et al., 2017]. There has been a recent surge in understanding the theoretical guarantees of Thompson Sampling due to its strong empirical evidence and simple implementation [Chapelle and Li, 2011]. In particular, for the stochastic multi-armed bandit problem, Agrawal and Goyal [2012] proved a problem-dependent logarithmic bound on expected regret of Thompson Sampling which was then showed to be asymptotically optimal [Kaufmann et al., 2012]. Subsequently, Agrawal and Goyal [2017] provided a problem-independent (i.e., worst-case) regret bound of $O(\sqrt{NT \log T})$ on the expected regret when using Beta priors. Interestingly, the expected regret can be improved to $O(\sqrt{NT \log N})$ by using Gaussian priors. Very recently, Jin et al. [2020] developed Minimax Optimal Thompson Sampling (MOTS), a variant of Thompson Sampling that achieves the minimax optimal regret of $O(\sqrt{NT})$. Agrawal and Goyal [2013b] also extended the analysis of multi-armed Thompson Sampling to the linear contextual setting and proved a regret bound of $\tilde{O}(d^{3/2}\sqrt{T})$ where $d$ is the dimension of the context vectors. In this paper, we develop the first variants of Batch Thompson Sampling that achieve the aforementioned regret bounds (problem-dependent and problem-independent versions) while reducing the sequential interaction with the environment from $T$ to $O(N \log T)$, thus increasing the efficiency of running Thompson Sampling by an exponential factor (for a fixed $N$).

There has been a large body of work and numerous algorithms for regret minimization of multi-armed bandit problems, including upper confidence bound (UCB), $\epsilon$-greedy, explore-then-commit, among many others. We refer the interested readers to some recent surveys for more details [Lattimore and Szepesvári, 2020, Slivkins, 2019]. The closest line of work to our paper is the proposed batch UCB algorithm [Gao et al., 2019], for which Esfandiari et al. [2021a] showed an asymptotically optimal regret bound with $O(\log T)$ number of batches. Very recently, Esfandiari et al. [2021a]

and Ruan et al. [2021] also addressed the batch linear bandits and the batch linear contextual bandits, respectively. Our work extends those results to the case of Thompson Sampling for the stochastic multi-armed bandit as well as the linear contextual bandit problems.

As we have highlighted in our proofs, our work builds on previous art, especially Agrawal and Goyal [2012, 2017, 2013b] (we believe that giving due credits to previous work is a virtue and not vice). However, we build on a non-trivial way. As it is clear from their analysis (and more generally for randomized probability matching strategies), breaking the sequential nature of distribution updates is non-trivial. We show that by a careful batch-mode strategy, one can reduce the sequential updates from $T$ to $O(\log(T))$. We are unaware of any previous work that obtains such a result for Thompson Sampling. In contrast, UCB strategies are much more amenable to parallelization (and the analysis is simple) as one can simply use the arm elimination method proposed by Esfandiari et al. [2021a] and Gu et al. [2021]. There is no clear way to use the arm elimination strategy for batch TS. Moreover, batch TS clearly outperforms the fully sequential UCB in all of our empirical results.

The benefits of batch-mode optimization has been considered in other machine learning settings, including convex optimization [Balkanski and Singer, 2018b, Chen et al., 2020], reinforcement learning [Zhang et al., 2020], submodular optimization [Chen et al., 2019, Fahrbach et al., 2019, Balkanski and Singer, 2018a], Gaussian processes [Desautels et al., 2014, Kathuria et al., 2016, Contal et al., 2013], stochastic sequential optimization [Esfandiari et al., 2021b, Agarwal et al., 2019, Chen and Krause, 2013], and Bayesian optimization [Wang et al., 2018, Rolland et al., 2018], to name a few.

Finally, we would also like to mention a concurrent work by Kalkanli and Ozgur [2021] on the very same subject of our paper, also accepted to NeurIPS 2021. Clearly, there is a significant overlap between the two works. For instance, in both papers, we have the problem-dependent asymptotic optimal regret with $O(\log T)$ batches. However, there are also a few differences. In particular, Kalkanli and Ozgur [2021] provide a tighter problem dependent bound on the *expected* number of batches for the stochastic multi-armed bandit. In contrast, we generalize the batch Thompson Sampling strategy to the linear contextual setting.

## 3 Preliminaries and Problem Formulation

As stated earlier, a standrad bandit problem is a repeated sequential game between a learner and the environment where at each round $t = 1, 2, \ldots, T$, the learner selects an action $a(t)$ from the set of actions $\mathcal{A}$ and then the environment reveals the reward $r_{a(t)} \in \mathbb{R}$. Different structures on the set of actions and rewards define different bandit problems. In this paper, we mainly consider two canonical variants, namely, stochastic multi-armed bandit, and stochastic linear contextual bandit.

**Stochastic Multi-Armed Bandit.** In this setting, the set of actions $\mathcal{A}$ is finite, namely, $\mathcal{A} = [N]$, and each action $a \in [N]$ is associated with a sub-Gaussian distribution $P_a$ (e.g., Bernoulli distribution, distributions supported on $[0, 1]$, etc). When the player selects an action $a$, a reward $r_a$ is sampled *independently* from $P_a$. We denote by $\mu_a = \mathbb{E}_{a \sim P_a}[r_a]$ the average reward of an action $a$ and by $\mu^* = \max_{a \in \mathcal{A}} \mathbb{E}_{a \sim P_a}[r_a]$ the action with the maximum average reward. Suppose the player selects actions $a_1, \ldots, a_T$ and receives the stochastic rewards $r_{a(1)}, \ldots, r_{a(T)}$. Then the (expected) regret is defined as

$$\mathcal{R}(T) = T\mu^* - \mathbb{E}\left[\sum_{t=1}^{T} r_{a(t)}\right].$$

We say that a policy achieves *no-regret*, if $\mathbb{E}[\mathcal{R}(T)]/T \to 0$ as the horizon $T$ tends to infinity. In order to compare the regret of algorithms, there are multiple choices in the literature. Once we fully specify the horizon $T$, the class of the bandit problem (e.g., multi-armed bandit with $N$ arms) and the specific instance we encounter withing the class (e.g., $\mu_1, \ldots, \mu_N$ in the stochastic multi-armed problem), then we can consider the *problem-dependent* regret bounds for each specific instance. In contrast, *problem-independent* bounds (also called worst-case bounds) only depends on the horizon $T$ and class of bandits for which the algorithm is designed (which is the number of arms $N$ in the multi-armed stochastic bandit problem), and not the specific instance within that class[1]. For the

---

[1]There is a related notion of regret, called *Bayesian regret*, considered in the Thompson Sampling literature [Russo and Van Roy, 2014, Bubeck and Liu, 2013], where a known prior on the environment is assumed. The

problem-dependent regret bound, it is known that UCB-like algorithms [Auer, 2002, Garivier and Cappé, 2011, Maillard et al., 2011] and Thomson Sampling [Agrawal and Goyal, 2013a, Kaufmann et al., 2012] achieve the asymptotic regret of $O(\log T \sum_{\Delta_a > 0} \Delta_a^{-1})$ where $\Delta_a = \mu^* - \mu_a \geq 0$. It is also known that no algorithm can achieve a better asymptotic regret bound [Lai and Robbins, 1985], thus implying that UCB and TS are both asymptotically optimal. In contrast, for the stochastic multi-armed bandit, UCB achieves the minimax problem-independent regret bound of $\sqrt{NT}$ [Auer, 2002] whereas TS (with Beta-priors) achieves a slightly worst regret of $\sqrt{NT \log T}$ [Agrawal and Goyal, 2017]. Very recently, Jin et al. [2020] developed Minimax Optimal Thompson Sampling (MOTS) that achieves the minimax optimal regret of $O(\sqrt{NT})$.

**Contextual Linear Bandit.** Contextual linear bandits generalise the multi-armed setting by allowing the learner to make use of side information. More specifically, each arm $a$ is associated with a feature/context vector $b_a \in \mathbb{R}^d$. At the beginning of each round $t \in [T]$, the learner first observes the contexts $b_a(t)$ for all $a \in \mathcal{A}$, and then she chooses an action $a(t) \in \mathcal{A}$. We assume that a feature vector $b_a$ affects the reward in a linear fashion, namely, $r_a(t) = \langle b_a(t), \mu \rangle + \eta_{a,t}$. Here, the parameter $\mu$ is unknown to the learner, and $\eta_{a,t}$ is an independent zero-mean sub-Gaussian noise given all the actions and rewards up to time $t$. Therefore, $\mathbb{E}[r_a(t)|b_a(t)] = \langle b_a(t), \mu \rangle$. The learner is trying to guess the correlation between $\mu$ and the contexts $b_a(t)$. For the set of actions $a(1), \ldots, a(T)$, the regret is defined as

$$\mathcal{R}(T) = \left[ \sum_{t=1}^{T} r_{a^*(t)}(t) \right] - \left[ \sum_{t=1}^{T} r_{a(t)}(t) \right],$$

where $a^*(t) = \arg\max_a \langle b_a(t), \mu \rangle$. The context vectors at time $t$ are generally chosen by an adversary after observing the actions played and the rewards received up to time $t - 1$. In order to obtain scale-free regret bounds, it is commonly assumed that $\|\mu\|_2 \leq 1$ and $\|b_a(t)\|_2 \leq 1$ for all arms $a \in \mathcal{A}$. By applying UCB to linear bandit, it is possible to achieve $\mathcal{R}(T) = \tilde{O}(d\sqrt{T})$ with high probability [Auer, 2002, Dani et al., 2008, Rusmevichientong and Tsitsiklis, 2010, Abbasi-Yadkori et al., 2011]. In contrast, Agrawal and Goyal [2013b] showed that the regret of Thompson Sampling can be bounded by $\tilde{O}(d^{3/2}\sqrt{T})$.

**Batch Bandit.** The focus of this paper is to parallelize the sequential decision making problem. In contrast to the fully sequential setting, where the learner selects an action and immediately receives the reward, in the batch mode setting, the learner selects a batch of actions and receives the rewards of all of them simultaneously (or only after the last action is executed). More formally, let the history $\mathcal{H}_t$ consists of all the actions and rewards up to time $t$, namely, $\{a(s)\}_{s \in [t-1]}$ and $\{r_{a(s)}(s)\}_{s \in [t-1]}$, respectively. We also denote the observed set of contexts up to and including time $t$ by $C_t = \{b_a(s)\}_{a \in \mathcal{A}, s \in [t]}$. Note that in the multi-armed bandit problem $C_t = \emptyset$. A fully sequential policy $\pi$ at round $t \in [T]$ maps the history and contexts to an action, namely, $\pi_t : \mathcal{H}_t \times C_t \to \mathcal{A}$. In contrast, a batch policy $\pi$ only interacts with the environment at rounds $0 = t_0 < t_1 < t_2 \cdots < t_m = T$. The $l$-th batch of duration $t_l - t_{l-1}$ contains the time units $\{t_{l-1} + 1, t_{l-1} + 2, \ldots, t_l\}$ which we denote it by the shorthand $(t_{l-1}, t_l]$. To select the actions in the $l$-th batch the policy is only allowed to use the history of actions/rewards observed in the previous batches, in addition to the contexts received so far. Therefore, a batch policy at time $t \in (t_{l-1}, t_l]$ is the following map: $\pi_t : \mathcal{H}_{t_{l-1}} \times C_t \to \mathcal{A}$. Moreover, a batch policy with a predetermined fixed batch size is called *static* and the one with a dynamic batch size is called *dynamic*.

## 4 Batch Thompson Sampling for Stochastic Multi-armed Bandit

In the classic Thompson Sampling (TS), at any time $t \in [T]$, we consider a prior distribution $D_a(t)$ on the underlying parameters of the reward distribution for every arm $a \in [N]$. TS works by first sampling $\theta_a(t) \sim D_a(t)$, independently for each $a \in [N]$, and then choosing the one with the highest value, namely, $a_t = \text{argmax}_{a \in [N]} \theta_a(t)$. Once the action $a_t$ is played, we receive the reward $r_t$, based on which the the prior distributions are updated as follows. If an arm $a$ is not selected, its distribution does not change, i.e., $D_a(t + 1) = D_a(t)$. However, if $a = a_t$, then we update $D_a(t + 1)$ given the information $(a_t, r_t)$ using the Bayes rule. By instantiating TS with different

---

frequentist regret bounds considered in this paper immediately imply a regret bound on the Bayesian regret but the opposite is not generally possible [Lattimore and Szepesvári, 2020].

| **Algorithm 1 Batch Thompson Sampling** | **Algorithm 2 Batch TS for Contextual Bandits** |
|---|---|
| 1: **Initialize:** $k_a \leftarrow 0$ ($\forall a \in [N]$), $l_a \leftarrow 0$ ($\forall a \in [N]$), batch $\leftarrow \emptyset$ | 1: **Initialize:** $k_a \leftarrow 0$ ($\forall a \in [N]$), $l_a \leftarrow 0$ ($\forall a \in [N]$), batch $\leftarrow \emptyset$, $B = I_d$, $\hat{\mu} = 0_d$ |
| 2: **for** $t = 1, 2, \cdots T$ **do** | 2: **for** $t = 1, 2, \cdots T$ **do** |
| 3: $\quad \theta_a(t) \sim D_a(t)$ ($\forall a \in [N]$) | 3: $\quad \tilde{\mu}(t) \sim \mathcal{N}(\hat{\mu}(B(t)), v^2 \mathcal{B}(B(t))^{-1})$ |
| 4: $\quad a(t) := \arg\max_{a \in [N]} \theta_a(t)$. | 4: $\quad a(t) = \arg\max_a b_a(t)^T \tilde{\mu}(t)$ |
| 5: $\quad k_{a(t)} \leftarrow k_{a(t)} + 1$ | 5: $\quad k_{a(t)} \leftarrow k_{a(t)} + 1$ |
| 6: $\quad$ **if** $k_{a(t)} < 2^{l_{a(t)}}$ **then** | 6: $\quad$ **if** $k_{a(t)} < 2^{l_{a(t)}}$ **then** |
| 7: $\qquad$ batch $\leftarrow$ batch $\cup \{a(t)\}$ | 7: $\qquad$ batch $\leftarrow$ batch $\cup \{a(t)\}$ |
| 8: $\quad$ **else** | 8: $\quad$ **else** |
| 9: $\qquad l_{a(t)} = l_{a(t)} + 1$ | 9: $\qquad l_{a(t)} = l_{a(t)} + 1$ |
| 10: $\qquad$ Query(batch) and receive rewards | 10: $\qquad$ Query(batch) and receive rewards |
| 11: $\qquad$ Update $D_a(t)$ ($\forall a \in$ batch) | 11: $\qquad$ Update $\hat{\mu}$ |
| 12: $\qquad$ batch $\leftarrow \emptyset$ | 12: $\qquad$ batch $\leftarrow \emptyset$ |
| 13: $\quad$ **end if** | 13: $\quad$ **end if** |
| 14: **end for** | 14: **end for** |

prior distributions (e.g., Beta, Gaussian), for which Bayes update is simple to compute, it is possible to show that one can achieve an asymptotically optimal regret [Agrawal and Goyal, 2012, 2017].

The main idea behind the Batch Thompson Sampling (B-TS), outlined in Algorithm 1, is as follows. For each arm $a \in [N]$, B-TS keeps track of $\{k_a\}_{a \in [N]}$, the number of times the arm $a$ has been selected so far. Initially, all $k_a$'s are set to 1. For each arm $a$ and at the beginning of the batch, necessarily $2^{l_a - 1} \leq k_a < 2^{l_a}$ for some integer $l_a \geq 1$. Now consider a new batch that starts at time $t$. Within this batch, B-TS samples arms according to the prior distributions up to time $t - 1$, namely $[D_a(t-1)]_{a \in [N]}$, and selects the one with the highest value. B-TS keeps selecting arms until the point that for one of the arms, say $a$, it reaches $k_a = 2^{l_a}$. At this point, B-TS queries all the arms selected during this batch. Based on the received rewards, B-TS updates $\{D_a\}_{a \in [N]}$ and starts a new batch. We should note that the doubling trick used in Algorithm 1 is reminiscent to the halving trick used in the pure exploration setting [Jun et al., 2021, Fiez et al., 2019].

**Regret Bounds with Beta Priors.** For the ease of presentation, we first consider the Bernoulli multi-armed bandit where $r_a \in \{0, 1\}$ and $\mu_a = \Pr[r_a = 1]$. In this setting, we can instantiate TS with Beta priors as follows. TS assumes an independent Beta-distributed prior, with parameters $(\alpha_a, \beta_a)$, over each $\mu_a$. Due to the nice conjugacy property of Beta distributions, it is very easy to update the posterior distribution, given the observations. In particular, the Bayes update can be performed as follows:

$$(\alpha_a, \beta_a) = \begin{cases} (\alpha_a, \beta_a) & \text{if } a(t) \neq a, \\ (\alpha_a, \beta_a) + (r(t), 1 - r(t)) & \text{if } a(t) = a. \end{cases}$$

TS initially assumes $\alpha_a = \beta_a = 1$ for all arms $a \in [N]$, which corresponds to the uniform distribution over $[0, 1]$. The update rule of B-TS in Algorithm 1 is also very similar. Let $B(t)$ be the last time $t' \leq t - 1$ that B-TS carried out a batch. Moreover, for each arm $a$, let $S_a(t)$ be the number of instances arm $a$ was selected by time $t - 1$ and $r_a = 1$. Similarly, let $F_a(t)$ be the number of instances arm $a$ was selected by time $t - 1$ and $r_a = 0$. We also denote by $k_a(t) = S_a(t) + F_a(t)$ the total number of instances arm $a$ was selected by time $t - 1$. Initially, B-TS starts with the uniform distribution over $[0, 1]$, i.e., $D_a(1) = Beta(1, 1)$ for all $a \in [N]$. Inspired by the update rule of TS, at any time $t$, B-TS updates the distribution $D_a(t)$ by $Beta(S_a(B(t)) + 1, F_a(B(t)) + 1)$. Note that during a batch when arms are being selected, the distributions $\{D_a\}_{a \in [N]}$ do not change. The updates only take place once the batch is carried out and the rewards are observed.

First we bound the number of batch queries as follows.

**Theorem 4.1.** *The total number of batches carried out by B-TS is at most $O(N \log T)$.*

The proof is given in Appendix A.2.

**Remark 4.2.** One might be tempted to show a sublinear dependency on $N$. However, simple empirical results show that the number of batches carried out by B-TS indeed scales logarithmically in $T$ but linearly in $N$. Please see figs 1a and 1b for more details.

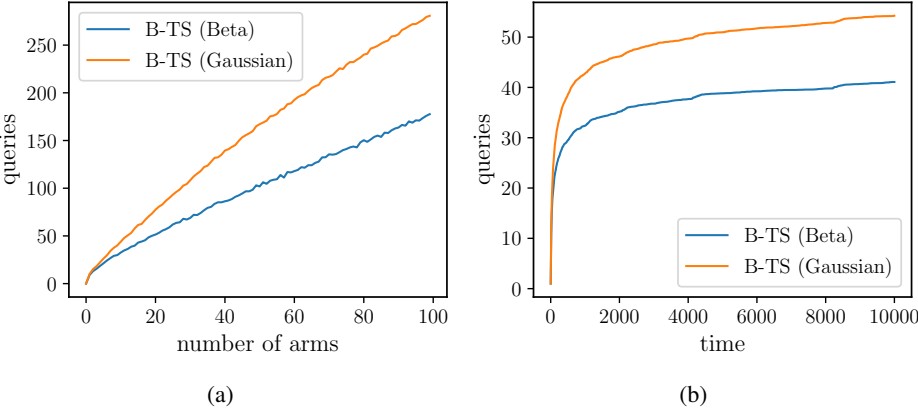

Figure 1: (a) and (b) show the number of batch queries versus the number of arms and the horizon, respectively. We consider a synthetic Bernoulli setting where the horizon is set to $T = 10^3$ and the number of arms vary from $N = 1$ to $N = 100$. We report the average regret over 100 experiments. As we clearly see in Figure 1a, the regret increases linearly in $N$ which rules out the possibility that the regret of B-TS may depend sub-linearly in $N$. For Figure 1b, we also consider the Bernoulli setting and set $N = 10$ and vary the horizon from $T = 1$ to $T = 10^4$. Again, as our theory suggests, the regret increases logarithmically in $T$.

What is more challenging is to show is that this simple batch strategy achieves the same asymptotic regret as a fully sequential one.

**Theorem 4.3.** *Without loss of generality, let us assume that the first arm has the highest mean value, i.e., $\mu^* = \mu_1$. Then, the expected regret of B-TS, outlined in Algorithm 1, with Beta priors can be bounded as follows*

$$\mathcal{R}(T) = (1 + \epsilon)O\left(\sum_{a=2}^{N} \frac{\ln T}{d(\mu_a, \mu_1)} \Delta_a\right) + O\left(\frac{N}{\epsilon^2}\right),$$

*where $d(\mu_a, \mu_1) := \mu_a \log \frac{\mu_a}{\mu_1} + (1 - \mu_a) \log \frac{(1-\mu_a)}{1-\mu_1}$ and $\Delta_a = \mu_1 - \mu_a$.*

The complete proof is given in Appendix A.2.

**Remark 4.4.** Even though we only provided the details for the Bernoulli setting, B-TS can be easily extended to general reward distributions supported over $[0, 1]$. To do so, once a reward $r_t \in [0, 1]$ is observed, we flip a coin with bias $r_t$ and update the Beta distribution according to the outcome of the coin. It is easy to see that Theorem 4.3 holds for this extension as well.

**Remark 4.5.** Gao et al. [2019] proved that for the $B$-batched $N$-armed bandit problem with time horizon $T$ it is necessary to have $B = \Omega(\log T / \log \log T)$ batches to achieve the problem-dependent asymptotic optimal regret. This lower bound implies that B-TS use almost the minimum number of batches needed (i.e., $O(\log T)$ versus $\Omega(\log T / \log \log T)$) to achieve the optimal regret.

Now we present the problem independent regret bound for B-TS.

**Theorem 4.6.** *Batch Thompson Sampling, outlined in Algorithm 1 and instantiated with Beta priors, achieves $\mathcal{R}(T) = O(\sqrt{NT \ln T})$ with $O(N \log T)$ batch queries.*

The proof is given in Appendix A.3.

**Regret Bounds with Gaussian Priors.** In order to obtain a better regret bound, we can instantiate TS with Gaussian distributions. To do so, let us define the empirical mean estimator for each arm $a$ as follows:

$$\hat{\mu}_a(t) = \frac{\sum_{\tau=1}^{t-1} r_{a(\tau)} \times \mathbb{I}(a(\tau) = a)}{k_a(t) + 1},$$

where $k_a(t)$ denotes the number of instances that an arm $a \in [N]$ has been selected up to time $t - 1$ and $\mathbb{I}(\cdot)$ is the indicator function. Then, by assuming that the prior distribution of an arm $a$ is

---

**Algorithm 3 Batch Minimax Optimal Thompson Sampling (B-MOTS)**

---
1: **Initialize:** $k_a \leftarrow 0 \ (\forall a \in [N])$, $l_a \leftarrow 0 \ (\forall a \in [N])$, batch $\leftarrow \emptyset$
2: **Initialize:** Play each arm $a$ once and initialize $D_a(t)$.
3: **for** $t = N + 1, \cdots T$ **do**
4:      for all arms $a \in [N]$ sample

$$\begin{aligned} \tilde{\theta}_a(t) &\sim \mathcal{N}(\hat{\mu}_a(B(t)), 1/(\rho k_a(B(t)))) \\ \theta_a(t) &\sim D_a(t) = \min\{\tilde{\theta}_a(t), \tau_a(t)\} \end{aligned}$$

5:      $a(t) := \operatorname{argmax}_a \theta_a(t)$
6:      $k_{a(t)} \leftarrow k_{a(t)} + 1$
7:      **if** $k_{a(t)} < 2^{l_{a(t)}}$ **then**
8:          batch $\leftarrow$ batch $\cup \{a(t)\}$
9:      **else**
10:          $l_{a(t)} = l_{a(t)} + 1$
11:          Query(batch) and observe rewards
12:          Update $D_a(t), \forall a \in [N]$
13:          batch $\leftarrow \emptyset$
14:      **end if**
15: **end for**

---

$\mathcal{N}\left(\hat{\mu}_a(t), \frac{1}{k_a(t)+1}\right)$, and that the likelihood of $r_{a_t}$ given $\mu_a$ is $\mathcal{N}(\mu_a, 1)$, the posterior will also be a Gaussian distribution with parameters $\mathcal{N}\left(\hat{\mu}_a(t+1), \frac{1}{k_a(t+1)+1}\right)$.

In B-TS, we need to slightly change the way we estimate $\hat{\mu}_a(t)$ as the algorithm has only access to the information received by the previous batches. Recall that $B(t)$ indicates the last time $t' \leq t - 1$ that B-TS carried out a batch query. For each arm $a \in [N]$, we assume the prior distribution $D_a(t) \sim \mathcal{N}\left(\hat{\mu}_a(B(t)), \frac{1}{k_a(B(t))+1}\right)$. We also update the empirical mean estimator as follows:

$$\hat{\mu}_a(t) = \frac{\sum_{\tau=1}^{B(t)} r_{a(\tau)} \times \mathbb{I}(a(\tau) = a)}{k_a(B(t) + 1) + 1}. \tag{1}$$

Note that at any time $t$ during the $l$-th batch, i.e., $t \in (t_{l-1}, t_l]$, the distribution $D_a(t)$ remains unchanged. Once the arms $\{a_t\}_{t \in (t_{l-1}, t_l]}$ are carried out and the rewards $\{r_t\}_{t \in (t_{l-1}, t_l]}$ are observed, $\hat{\mu}_a$ changes and the posterior is computed accordingly, namely, $D_a(t_l + 1) \sim \mathcal{N}\left(\hat{\mu}_a(t_l + 1), \frac{1}{k_a(t_l+1)+1}\right)$. As we instantiate B-TS with Gaussian priors, the regret bound slightly improves.

**Theorem 4.7.** *Batch Thompson Sampling, outlined in Algorithm 1 and instantiated with Gaussian priors, achieves* $\mathbb{E}[\mathcal{R}(T)] = O(\sqrt{NT \ln N})$ *with* $O(N \log T)$ *batch queries.*

The proof is given in Appendix A.4.

## 5   Batch Minimax Optimal Thompson Sampling

So far, we have considered the parallelization of the vanilla Thompson Sampling which does not achieve the optimal minimax regret. In this section, we introduce Batch Minimax Optimal Thompson Sampling (B-MOTS), that achieves the optimal minimax bound of $O(\sqrt{NT})$, as well as the asymptotic optimal regret bound for Gaussian rewards. In contrast to the fully sequential MOTS developed by Jin et al. [2020], B-MOTS requires only $O(N \log T)$ batches. The crucial difference between B-MOTS and B-TS is that instead of choosing Gaussian or Beta distributions, B-MOTS uses a clipped Gaussian distribution.

To run B-MOTS, we need to slightly change the way $D_a(t)$ is updated. First, to initialize $D_a(t)$, B-MOTS plays each arm once in the beginning and sets $k_a(N + 1)$ to 1 and $\hat{\mu}_a(N + 1)$ to the

observed reward of each arm $a \in [N]$. To determine $D_a(t)$ for the subsequent batches, let us first define a confidence range $(-\infty, \tau_a(t))$ for each arm $a \in [N]$ as follows:

$$\tau_a(t) = \hat{\mu}_a(B(t)) + \sqrt{\frac{\alpha}{k_a(B(t))} \log^+ \left( \frac{T}{N k_a(B(t))} \right)}, \tag{2}$$

where $\log^+(x) = \max\{0, \log(x)\}$ and the empirical mean for each arm $a$ is estimated as

$$\hat{\mu}_a(t) = \frac{\sum_{\tau=1}^{B(t)} r_{a(\tau)} \times \mathbb{I}(a(\tau) = a)}{k_a(B(t) + 1)}. \tag{3}$$

Note that the estimators in (3) and (1) slightly differ due to the initialization step of B-MOTS.

For each arm $a \in [n]$, B-MOTS first samples $\tilde{\theta}_a(t)$ from a Gaussian distribution with the following parameters $\tilde{\theta}_a(t) \sim \mathcal{N}(\hat{\mu}_a(B(t)), 1/(\rho k_a(B(t))))$, where $\rho \in (1/2, 1)$ is a tuning parameter. Then, the sample is clipped by the confidence range as follows:

$$D_a(t) = \min\{\tilde{\theta}_a(t), \tau_a(t)\}. \tag{4}$$

The rest is exactly as in Alg 1 for B-TS. If you are interested in the details, you can find the outline of B-MOTS algorithm in Appendix B.

**B-MOTS for SubGaussian Rewards.** We state the regret bounds in the most general format, i.e., when the rewards follow a sub-Gaussian distribution. To remind ourselves, we say that a random variable $X$ is $\sigma$ sub-Gaussian if $\mathbb{E}[\exp(\lambda X - \lambda \mathbb{E}[X])] \le \exp(\sigma^2 \lambda^2 / 2)$, for all $\lambda \in \mathbb{R}$.

The following theorem shows that B-MOTS is minimax optimal.

**Theorem 5.1.** *If the reward of each arm is 1-subgussian then the regret of B-MOTS is bounded by* $\mathcal{R}(T) = O(\sqrt{NT} + \sum_{a:\Delta_a>0} \Delta_a)$. *Moreover, the number of batches is bounded by $O(N \log T)$.*

The proof is given in Appendix B.2.

The next theorem presents the asymptotic regret bound of B-MOTS for sub-Gaussian rewards.

**Theorem 5.2.** *Assume that the reward of each arm $a \in [N]$ is 1-subgussian with mean $\mu_a$. For any fixed $\rho \in (1/2, 1)$, the regret of B-MOTS can be bounded as $\mathcal{R}(T) = O\left( \log(T) \sum_{a:\Delta_a>0} \frac{1}{\rho \Delta_a} \right)$.*

The proof is given in Appendix B.3

The asymptotic regret rate of B-MOTS matches the existing lower bound $\log(T) \sum_{a:\Delta_a>0} 1/\Delta_a$ [Lai and Robbins, 1985] up to a multiplicative factor $1/\rho$. Therefore, similar to the analysis of the fully sequential setting [Jin et al., 2020], B-MOTS reaches the exact lower bound at a cost of minimax optimality. In the next section, we show that at least in the Gaussian reward setting, minimax and asymptotic optimally cab be achieved simultaneously.

**B-MOTS-J for Gaussian Rewards.** In this part we present a batch version of Minimax Optimal Thompson Sampling for Gaussian rewards [Jin et al., 2020], called B-MOTS-J, which achieves both minimax and asymptotic optimality when the reward distribution is Gaussian. The only difference between B-MOTS-J and B-MOTS is the way $\tilde{\theta}_a(t)$ are sampled. In particular, B-MOTS-J samples $\tilde{\theta}_a(t)$ according to $\mathcal{J}(\mu, \sigma^2)$ (instead of a Gaussian distribution), where the PDF is defined as

$$\Phi_{\mathcal{J}}(x) = \frac{1}{2\sigma^2} |x - \mu| \exp \left[ -\frac{1}{2} \left( \frac{x - \mu}{\sigma} \right)^2 \right].$$

Note that when $x$ is restricted to $x \ge 0$, then $\mathcal{J}$ becomes a Rayleigh distribution. More precisely, to sample $\tilde{\theta}_a(t)$, we set the parameters of $\mathcal{J}$ as follows: $\tilde{\theta}_a(t) \sim \mathcal{J}\left( \hat{\mu}_a(B(t)), \frac{1}{k_a(B(t))} \right)$, where $\hat{\mu}_a(t))$ is estimated according to (3). The rest of the algorithm is run exactly like B-MOTS.

**Theorem 5.3.** *Assume that the reward of each arm $a$ is sampled from a Gaussian distribution $\mathcal{N}(\mu_a, 1)$ and $\alpha > 2$. Then, the regret of B-MOTS-J can be bounded as follows:*

$$\mathcal{R}(T) = O(\sqrt{KT} + \sum_{a=2}^{k} \Delta_a), \quad \lim_{T \to \infty} \frac{\mathcal{R}(T)}{\log(T)} = \sum_{a:\Delta_a>0} \frac{2}{\Delta_a}.$$

The proof is given in Appendix B.4.

## 6 Batch Thompson Sampling for Contextual Bandits

In this section, we propose Batch Thompson Sampling for Contextual Bandits (B-TS-C), outlined in Algorithm 2. As in the fully sequential TS, proposed by Agrawal and Goyal [2013b], we assume Gaussian priors and Gaussian likelihood functions. However, we should highlight that the analysis of B-TS-C and the corresponding regret bound hold irrespective of whether or not the reward distribution matches the Gaussian priors and Gaussian likelihood functions (similar to the multi-armed bandit setting discussed in Section 4). More formally, given a context $b_a(t)$, and parameter $\mu$, we assume that the likelihood of the reward $r_a(t)$ is given by $\mathcal{N}(b_a(t)^T\mu, v^2)$, where $v = \sigma\sqrt{9d\ln(T/\delta)}$ and $\delta \in (0,1)^2$. Let us define the matrix $\mathcal{B}(t)$ as follows

$$\mathcal{B}(t) = I_d + \sum_{\tau=1}^{t-1} b_{a(\tau)}(\tau)b_{a(\tau)}(\tau)^T.$$

Note that the matrix $\mathcal{B}(t)$ depends on all the contexts observed up to time $t-1$. We consider the prior $\mathcal{N}(\hat{\mu}(B(t)), v^2\mathcal{B}(B(t))^{-1})$ for $\mu$ and update the the empirical mean estimator as follows:

$$\hat{\mu}(t) = \mathcal{B}(B(t))^{-1}\left(\sum_{\tau=1}^{B(t)} b_{a(\tau)}(\tau) \times r_{a(\tau)}(\tau)\right). \tag{5}$$

Note that in order to estimate $\hat{\mu}(t)$, we only consider the rewards received up to time $B(t)$, namely, the rewards of arms pulled in the previous batches. At each time step $t$, B-TS-C generates a sample $\tilde{\mu}(t)$ from $\mathcal{N}(\hat{\mu}(B(t)), v^2\mathcal{B}(B(t))^{-1})$ and plays the arm $a$ that maximizes $b_a(t)^T\tilde{\mu}(t)$. The posterior distribution for $\mu$ at time $t+1$ will be $\mathcal{N}(\hat{\mu}(B(t+1)), v^2\mathcal{B}(B(t+1))^{-1})$.

**Theorem 6.1.** *The B-TS-C algorithm (Algorithm 2) achieves the total regret of*

$$\mathcal{R}(T) = O\left(d^{3/2}\sqrt{T}(\ln(T) + \sqrt{\ln(T)\ln(1/\delta)})\right)$$

*with probability* $1 - \delta$. *Moreover, B-TS-C carries out* $O(N\log T)$ *batch queries.*

The proof is given in Appendix C.2.

## 7 Experimental Results

In this section, we compare the performance of our proposed batch Thompson Sampling policies (e.g., B-TS,B-MOTS, B-MOT-J and B-TS-C) with their fully sequential counterparts. We also include several baselines such as UCB and Thompson Sampling with static batch design (Static-TS). In particular, for Static-TS, Static-TS2, and Static-TS4, we set the total number of batches to that of B-TS, twice of B-TS, and four times of B-TS, respectively. However, in the static batch design, we use equal sized batches.

**Batch Thompson Sampling.** In Figure 2a, we compare the performance of UCB against TS and its batch variants in a synthetic Bernoulli setting. We vary $T$ from 1 to $10^4$ and set $N = 10$. We run all the experiments 1000 times. Figure 2a compares the average regret, i.e., $\mathcal{R}(T)/T$, versus the horizon $T$. As expected, TS outperforms UCB. Moreover, TS and B-TS follow the same trajectory and have practically the same regret. Note that for the static variant of TS, namely, Static-TS, we see that in the first few hundred iterations, its performance is even worst than UCB and then it catches with TS. Figure 2b more clearly shows the trade-off between the regret obtained by different baselines versus the number of batch queries (bottom-left is the desirable location). In this figure, we set $T = 10^3$. We see that the lowest regret is achieved by TS and B-TS but TS carries out many more queries. Also, we should highlight that while the static versions make fewer queries than TS, they do not achieve a similar regret. Notably, even Static-TS-4, that carries out 4 times more queries than B-TS, has a much higher regret. This shows the importance of a dynamic batch design.

---

[2]If the horizon $T$ is unknown, we can use $v_t = \sigma\sqrt{9d\ln(t/\delta)}$.

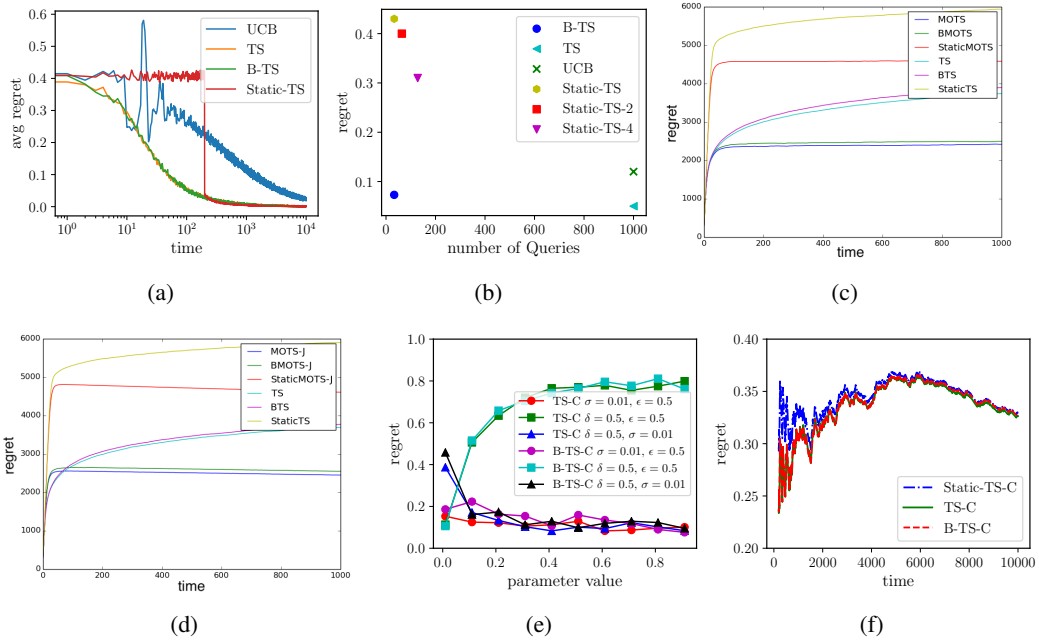

Figure 2: (a) and (b) compare the regret of UCB against TS and its batch variants. (c) and (d) compare the batch variants of TS and MOTS. (e) shows the sensitivity of TS-C and its batch variants to the tuning parameters. (f) shows the performance of TS-C and its batch variants on real data.

**Batch Minimax Optimal Thompson Sampling.**    In Figures 2c and 2d we compare the regret of MOTS Jin et al. [2020] and its batch versions. The synthetic setting is similar to Jin et al. [2020]. We set $N = 50$, $T = 10^6$ and the total number of runs to 2000. The reward of each arm is sampled from an independent Gaussian distribution. More precisely, the optimal arm has the expected reward and variance 1 while the other $N - 1$ arms have the expected reward $1 - \epsilon$ and variance 1 (we set $\epsilon = 0.2$). For MOTS, we set $\rho = 0.9999$ and $\alpha = 2$ as suggested by Jin et al. [2020]. As we can see in Figure 2c, the batch variants of TS and MOTS achieve practically a similar regret. Also, as our theory suggests, B-MOTS (along with MOTS) have the lowest regret while B-MOTS drastically reduces the number of batches w.r.t MOTS. Moreover, the static batch designs, namely Static-TS and Static-MOTS, show the highest regret while carrying out the same number of batch queries as B-TS and B-MOTS. Therefore, the dynamic batch design of B-TS and B-MOTS seems crucial for obtaining good performances. A similar trend is shown in Figure 2d where we run MOTS-J (with $\alpha = 2$) and its batch variants. Again, B-MOTS-J and MOTS-J are practically indistinguishable while achieving the lowest regret.

**Contextual Bandit.**    For the contextual bandit, we perform a synthetic and a real-data experiment. Figure 2e shows the performance of the sequential Thompson Sampling, namely, TS-C, and the batch variants, namely, B-TS-C, as we change different parameters $\epsilon, \delta$ and $\sigma$ from 0 to 1. Here, the context dimension is 5 and we set the horizon to $T = 10^4$. We run all the experiments 1000 times. As we see in Figure 2e, TS-C and B-TS-C follow practically the same curves.

For the experiment on real data, we use the MovieLens data set where the dimension of the context is 20, the horizon is $T = 10^5$, and we run each experiment 100 times. For the parameters, we set $\delta = 0.61, \sigma = 0.01$, and $\epsilon = 0.71$ as suggested by Beygelzimer et al. [2011]. We see that Static-TS-C performs a bit worst than TS-C and B-TS-C, again suggesting that it is crucial to use dynamic batch sizes.

## 8    Conclusion

In this paper, we revisited the classic Thompson Sampling procedure and developed the first Batch variants for the stochastic multi-armed bandit and linear contextual bandit. We proved that our proposed batch policies achieve similar regret bounds (up to constant factors) but with significantly fewer number of interactions with the environment. We have also demonstrated experimentally that our batch policies achieve practically the same regret on both synthetic and real data.

## Funding Transparency Statement

This research is supported by NSF CAREER Award (IIS-1845032), ONR YIA (N00014-19-1-2406), and TATA.

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
