# Appendices

## A  Batch Thompson Sampling for Multi-armed Bandit

In this section, we follow the notations used in Agrawal and Goyal [2012, 2017] and adapt them to the batch setting.

### A.1  Notations and Definitions

**Definition A.1.** For a Binomial distribution with parameters $\alpha$ and $\beta$, we refer to its CDF as $F_{n,p}^B(.)$, and pdf as $f_{n,p}^B(.)$. We furthermore denote by $F_{\alpha,\beta}^{beta}(.)$ the CDF of Beta distribution. It is easy to show that for all $\alpha, \beta > 0$,
$$F_{\alpha,\beta}^{beta}(y) = 1 - F_{\alpha+\beta-1,y}^B(\alpha - 1) .$$

**Definition A.2** (History/filtration $\mathcal{F}_t$)**.** For time steps $t = 1, \cdots, T$, we define $\mathcal{F}_t$ as the history of the arms that have been played upto time $t$, i.e.,
$$\mathcal{F}_t = \{a(\tau), r_{a(t)}(\tau), \tau \leq t\} .$$

**Definition A.3.** For a given arm $a$, we denote by $\tau_j$ the time step in which $a$ has been *queried* for the $j$-th time. We let $\tau_0 = 0$. Note that $\tau_T \geq T$.

**Definition A.4.** We denote by $\theta_a(t)$ the sample for arm $a$ at time $t$ from the posterior distribution at time $B(t)$, namely $Beta(S_a(B(t)) + 1, k_a(B(t)) - S_a(B(t)) + 1)$.

**Definition A.5.** Without loss of generality, we assume that $a = 1$ is the optimal arm. For a non-optimal arm $a \neq 1$, we have two thresholds $x_a, y_a$ depending on the type of upper bounds we are proving (i.e., problem dependent or independent) such that $\mu_a < x_a < y_a < \mu_1$.

**Definition A.6.** We denote by $\Delta_a' := \mu_1 - y_a$ and $D_a := y_a \ln \frac{y_a}{\mu_1} + (1 - y_a) \ln \frac{1-y_a}{1-\mu_1}$. Also define $d(\mu_a, \mu_1) := \mu \log \frac{\mu_a}{\mu_1} + (1 - \mu_a) \log \frac{1-\mu_a}{1-\mu_1}$.

**Definition A.7.** For a non-optimal arm $a$ (i.e., $a \neq 1$), we use $E_a^\mu(t)$ for the event $\{\hat{\mu}_a(B(t)) \leq x_a\}$ and we use $E_a^\theta(t)$ for the event $\{\theta_a(t) \leq y_a\}$.

**Definition A.8.** The (conditional) probability that for a non-optimal arm $a$, the generated sample for the optimal arm $a = 1$ at time $t$ exceeds the threshold $y_a$ is defined as
$$p_{a,t} := \Pr(\theta_1(t) > y_a | \mathcal{F}_{B(t)}) .$$

Here is our first lemma regarding the relationship between batch bandit and sequential bandit.

**Lemma A.9.** *For any arm $a$, we have $k_a(B(t)) \geq \frac{1}{2} k_a(t)$.*

*Proof.* The reason is that if $k_a(B(t)) < \frac{1}{2} k_a(t)$ then B-TS (Algorithm 1) should have queried a batch after time $B(t)$ which is a contradiction. $\square$

### A.2  Problem-dependent Regret Bound with Beta Priors

**Theorem 4.1.** *The total number of batches carried out by B-TS is at most $O(N \log T)$.*

*Proof.* Every time we query a batch, there is one arm $a$, for which $k_a = 2^{\ell_a}$. In order to count the total number of batches, we assign each time step $t$ to a batch $B$. Note that the assigned batch for $t$ is not necessarily the batch that $a(t)$ will be added to. Suppose $k_a = 2^{\ell_a}$, and the algorithm queries a batch $B$, we assign time steps in which arm $a$ was queried for the $2^{\ell_a - 1} + 1, \cdots, 2^{\ell_a}$-th times to the batch $B$ (although some of the elements might have been queried in the previous batches). Let's denote this set by $T_a(B)$. Then for each arm $a$, the total number of batches corresponding to arm $a$ is at most $O(\log T)$ (since the last time step arm $a$ is being played is at most $T$). Therefore, we can upper bound the total number of batches by $O(N \log T)$ batches. $\square$

First, note that in the batch algorithm B-TS (Algorithm 1), we define $\theta_a(t)$ based on $\mathcal{F}_{B(t)}$. As a result of these modifications the following lemma is immediate. It is a batch variation of [Agrawal and Goyal, 2017, Lemma 2.8].

**Lemma A.10.** *For all t,all suboptimal arm $a \neq 1$, and all instantiation $\mathcal{F}_{B(t)}$ we have*

$$\Pr(a(t) = a, E_a^\mu(t), E_a^\theta(t)|\mathcal{F}_{B(t)}) \leq \frac{1 - p_{a,t}}{p_{a,t}} \Pr(a(t) = 1, E_a^\mu(t), E_a^\theta(t)|\mathcal{F}_{B(t)}) \,.$$

*Proof.* $E_a^\mu(t)$ is determined by $\mathcal{F}_{B(t)}$. Therefore it is enough to show that for any instantiation $\mathcal{F}_{B(t)}$

$$\Pr(a(t) = a|E_a^\theta(t), \mathcal{F}_{B(t)}) \leq \frac{1 - p_{a,t}}{p_{a,t}} \Pr(a(t) = 1|E_a^\theta(t), \mathcal{F}_{B(t)}) \,.$$

Now given $E_a^\theta(t)$, we have $a(t) = a$ only if $\theta_j(t) \leq y_a, \forall j$. Therefore, for $a \neq 1$ and any instantiation $\mathcal{F}_{B(t)}$ we have

$$
\begin{aligned}
\Pr(a(t) = a|E_a^\theta(t), \mathcal{F}_{B(t)}) &\leq \Pr(\theta_j(t) \leq y_a, \forall j|E_a^\theta(t), \mathcal{F}_{B(t)}) \\
&= \Pr(\theta_1(t) \leq y_a|\mathcal{F}_{B(t)}) . \Pr(\theta_j(t) \leq y_a, \forall j \neq 1|E_a^\theta(t), \mathcal{F}_{B(t)}) \\
&= (1 - p_{a,t}) . \Pr(\theta_j(t) \leq y_a, \forall j \neq 1|E_a^\theta(t), \mathcal{F}_{B(t)}) \,.
\end{aligned}
$$

In the first equality given $\mathcal{F}_{B(t)}$, the random variable $\theta_1(t)$ is independent of all other $\theta_j(t)$ and $E_a^\theta(t)$. The argument for $a = 1$ is similar. $\qquad\square$

Now we prove the main lemma which provides a problem-dependent upper bound on the regret.

**Theorem 4.3.** *Without loss of generality, let us assume that the first arm has the highest mean value, i.e., $\mu^* = \mu_1$. Then, the expected regret of B-TS, outlined in Algorithm 1, with Beta priors can be bounded as follows*

$$\mathcal{R}(T) = (1 + \epsilon)O\left(\sum_{a=2}^N \frac{\ln T}{d(\mu_a, \mu_1)} \Delta_a\right) + O\left(\frac{N}{\epsilon^2}\right),$$

*where $d(\mu_a, \mu_1) := \mu_a \log \frac{\mu_a}{\mu_1} + (1 - \mu_a) \log \frac{(1 - \mu_a)}{1 - \mu_1}$ and $\Delta_a = \mu_1 - \mu_a$.*

*Proof.* The proof closely follows [Agrawal and Goyal, 2017, Theorem 1.1] and is adapted to the batch setting. For a non optimal arm $a \neq 1$, we decompose the expected number of plays of arm $a$ as follows

$$
\begin{aligned}
\mathbb{E}[k_a(t)] &= \sum_{t=1}^T \Pr(a(t) = a) \\
&= \sum_{t=1}^T \Pr(a(t) = a, E_a^\mu(t), E_a^\theta(t)) + \sum_{t=1}^T \Pr(a(t) = a, E_a^\mu(t), \overline{E_a^\theta(t)}) + \sum_{t=1}^T \Pr(a(t) = a, \overline{E_a^\mu(t)}) \,.
\end{aligned}
$$

$$(6)$$

The first term can be bounded by lemma A.10 as follows:

$$
\begin{aligned}
\sum_{t=1}^T \Pr(a(t) = a, E_a^\mu(t), E_a^\theta(t)) &\leq \sum_{t=1}^T \mathbb{E}\left[\Pr(a(t) = a, E_a^\mu(t), E_a^\theta(t)|\mathcal{F}_{B(t)})\right] \\
&\leq \sum_{t=1}^T \mathbb{E}\left[\frac{(1 - p_{a,t})}{p_{a,t}} \Pr(a(t) = 1, E_a^\theta(t), E_a^\mu(t))|\mathcal{F}_{B(t)}\right] \\
&= \sum_{t=1}^T \mathbb{E}\left[\mathbb{E}\left[\frac{1 - p_{a,t}}{p_{a,t}} \mathbb{I}(a(t) = 1, E_a^\theta(t), E_a^\mu(t))|\mathcal{F}_{B(t)}\right]\right] \\
&\leq \sum_{t=1}^T \mathbb{E}\left[\frac{1 - p_{a,t}}{p_{a,t}} \mathbb{I}(a(t) = 1, E_a^\theta(t), E_a^\mu(t))\right] \,.
\end{aligned}
$$

Note that as before, given $\mathcal{F}_{B(t)}$, the probability $p_{a,t}$ is fixed which implies the second inequality. The difference between this argument and that of the proof closely follows [Agrawal and Goyal, 2017, Theorem 1.1] is that conditioning is until the last time the B-TS algorithm has queried a batch, i.e., $B(t)$. Note that $p_{a,t} = \Pr(\{\theta_1(t) > y_a | \mathcal{F}_{B(t)}\})$ changes only after a batch queries the optimal arm. Hence as before $p_{a,t}$ remains the same at all time steps $t \in \{\tau_k + 1, \cdots, \tau_{k+1}\}$ (refer to Definition A.3). Thus we can get the following decomposition

$$\sum_{t=1}^{T} \mathbb{E}\left[\frac{1 - p_{a,t}}{p_{a,t}}\mathbb{I}(a(t) = 1, E_a^\theta(t), E_a^\mu(t))\right] \leq \sum_{k=0}^{T-1} \mathbb{E}\left[\frac{(1 - p_{a,\tau_k+1})}{p_{a,\tau_k+1}} \sum_{t=\tau_k+1}^{\tau_{k+1}} \mathbb{I}(a(t) = 1, E_a^\theta(t), E_a^\mu(t))\right]$$

$$\leq \sum_{k=0}^{T-1} \mathbb{E}\left[\frac{1 - p_{a,\tau_k+1}}{p_{a,\tau_k+1}}\right]. \tag{7}$$

Now for the term $\mathbb{E}\left[\frac{1}{p_{a,\tau_k+1}}\right]$, since $k_a(B(t)) \geq 1/2 k_a(t)$ (Lemma A.9), we can get a modification of the bound provided in Agrawal and Goyal [2017, Lemma 2.9], as follows.

**Lemma A.11.** *Let $\tau_k$ be the time step that optimal arm 1 has been played for the $k$-th time, Then for non optimal arm $a \neq 1$ we have,*

$$\mathbb{E}\left[\frac{1}{p_{a,\tau_k+1}} - 1\right] = \begin{cases} \frac{3}{\Delta_a'}, & \text{for } k < \frac{16}{\Delta_a'}, \\ \Theta\left(\exp(-\Delta_a'^2 k/4) + \frac{\exp(-D_a k/2)}{(k/2+1)\Delta_a^2} + \frac{1}{\exp(\Delta_a'^2 k/16) - 1}\right), & \text{otherwise.} \end{cases}$$

Similar to Agrawal and Goyal [2017, Lemma 2.10], we obtain the following lemma.

**Lemma A.12.** *For a non optimal arm $a \neq 1$, we have*

$$\sum_{t=1}^{T} \Pr(a(t) = a, E_a^\mu(t), E_a^\theta(t)) \leq \frac{48}{\Delta_a'^2} + \sum_{j > 16/\Delta_a'} \Theta\left(e^{-\Delta_a'^2 j/4} + \frac{2}{(j+1)\Delta_a'^2}\right)e^{-D_a j/2} + \frac{1}{e^{\Delta_a'^2 j/8} - 1}.$$

Now by substituting the above lemma into equation 7, we can upper bound other terms in equation (6) to prove the following lemma.

**Lemma A.13.** *For a non optimal arm $a \neq 1$, we have*

$$\sum_{t=1}^{T} \Pr(a(t) = a, \overline{E_a^\mu(t)}) \leq \frac{2}{d(x_a, \mu_a)} + 1.$$

*Proof.* Let $\tau_k$ be the $k$-th play of arm $a$. The LHS can be upper bounded by $\sum_{k=0}^{T-1} \Pr(\overline{E_a^\mu(\tau_{k+1})})$. Note that $\hat{\mu}_a$ will be updated when the algorithm queries a batch. Using Chernoff-Hoeffding bound

$$\Pr(\hat{\mu}_a(B(\tau_{k+1})) > x_a) \leq e^{-\frac{1}{2}kd(x_a, \mu_a)},$$

where $x_a$ is defined in Definition A.5. Note that at time $B(\tau_{k+1})$, arm $a$ has been played at least $k/2$ times. Thus,

$$\sum_{t=1}^{T} \Pr(\overline{E_a^\mu(\tau_{k+1})}) = \sum_{k=0}^{T-1} \Pr(\hat{\mu}_a(B(\tau_{k+1})) > x_a) \leq 1 + \sum_{k=1}^{T-1} \exp(-\frac{1}{2}kd(x_a, \mu_a)) \leq 1 + \frac{2}{d(x_a, \mu_a)}.$$

$\square$

The statement of the following lemma is similar to [Agrawal and Goyal, 2017, Lemma 2.12]. However, we prove it for the batch policy.

**Lemma A.14.** *For a non optimal arm $a \neq 1$, we have*

$$\sum_{t=1}^{T} \Pr(a(t) = a, \overline{E_a^\theta(t)}, E_a^\mu(t)) \leq L_a(t) + 1,$$

*where $L_a(t) = \frac{\ln T}{d(x_a, y_a)}$.*

*Proof.* We can consider two cases when $k_a(B(t))$ is large (greater than $L_a(t)$) or small (less than $L_a(t)$). This way, we have

$$\sum_{t=1}^{T} \Pr(a(t) = a, \overline{E_a^\theta(t)}, E_a^\mu(t)) = \sum_{t=1}^{T} \Pr(a(t) = a, k_a(B(t)) \leq L_a(t), \overline{E_a^\theta(t)}, E_a^\mu(t))$$

$$+ \sum_{t=1}^{T} \Pr(a(t) = a, k_a(B(t)) > L_a(t), \overline{E_a^\theta(t)}, E_a^\mu(t)). \quad (8)$$

Same as before the first term is bounded by $\mathbb{E}\left[\sum_{t=1}^{T} \mathbb{I}(a(t) = a, k_a(B(t)) \leq L_a(t))\right]$ which is bounded by $L_a(t)$. Again we bound the second term by 1. The main idea is to show that for large enough $k_a(B(t))$, and given $E_a^\mu(t)$ is true, the probability of $E_a^\theta(t)$ being false is small. We can write

$$\sum_{t=1}^{T} \Pr(a(t) = a, k_a(B(t)) > L_a(t), \overline{E_a^\theta(t)}, E_a^\mu(t)) = \mathbb{E}\left[\sum_{t=1}^{T} \mathbb{I}(k_a(t) > L_a(t), E_a^\mu(t)) \Pr(a(t) = a, \overline{E_a^\theta}(t) | \mathcal{F}_{B(t)})\right]$$

$$\leq \mathbb{E}\left[\sum_{t=1}^{T} \mathbb{I}(k_a(t) > L_a(t), \mu_a(B(\hat{t})) \leq x_a) \Pr(\theta_a(t) > y_a | \mathcal{F}_{B(t)})\right].$$

Note that $\mathcal{F}_{B(t)}$ determines both $k_a(B(t))$ and $E_a^\mu(t)$. Now, $\theta_a(t)$ is distributed according to

$$\theta_a(t) \sim Beta(\hat{\mu}_a(B(t))k_a(B(t) + 1, (1 - \hat{\mu}_a(B(t))k_a(B(t)))).$$

Given $E_a^\mu(t)$, it is stochastically dominated by $Beta(x_a k_a(B(t)) + 1, (1 - x_a)k_a(B(t)))$. Now, if $\mathcal{F}_{B(t)}$ contains the events $E_a^\mu(t)$ and $\{k_a(B(t)) > L_a(t)\}$, we have

$$\Pr(\theta_a(t) > y_a | \mathcal{F}_{B(t)}) \leq 1 - F_{x_a k_a(B(t))+1, (1-x_a)k_a(B(t))}^{beta}(y_a).$$

Using the Chernouf-Hoefding inequality, we can show that the RHS of the above inequality is at most

$$1 - F_{x_a k_a(B(t))+1, (1-x_a)k_a(B(t))}^{beta}(y_a) = F_{k_a(B(t))+1, y_a}^{B}(x_a(k_a(t) + 1))$$

$$\leq \exp(-(k_a(B(t)) + 1)d(x_a, y_a))$$

$$\leq \exp(-(L_a(t))d(x_a, y_a))$$

$$\leq 1/T.$$

Summing over $t$ yields the upper bound 1 for the second term in 8. $\qquad\square$

The rest of the proof is by combining the above lemmas and by setting the right value for $x_a$ and $y_a$ as discussed in Agrawal and Goyal [2017]. In particular, by combining Lemma A.12, A.13, and A.14 we have

$$\mathbb{E}[k_a(t)] \leq \frac{48}{\Delta_a'^2} + \sum_{j > 16/\Delta_a'} \Theta(e^{-\Delta_a'^2 j/4} + \frac{2}{(j+1)\Delta_a'^2})e^{-(D_a)j/2} + \frac{1}{e^{\Delta_a'^2 j/8} - 1}) + L_a(t) + 1 + \frac{1}{d(x_a, \mu_a)} + 1.$$

Now we should set the right value to parameters $x_a, y_a$. For $0 \leq \epsilon < 1$, set $x_a \in (\mu_a, \mu_1)$ such that $d(x_a, \mu_1) = d(\mu_a, \mu_1)/(1 + \epsilon)$ and set $y_a \in (x_a, \mu_1)$ such that $d(x_a, y_a) = d(x_a, \mu_1)/(1 + \epsilon) = d(\mu_a, \mu_1)/(1 + \epsilon)^2$. For these values, the regret bound easily follows. We will use different values for problem independent case in the next section.

$\qquad\square$

## A.3 Problem-independent Regret Bound with Beta Priors

Now we prove the problem independent regret bound.

**Theorem 4.6.** *Batch Thompson Sampling, outlined in Algorithm 1 and instantiated with Beta priors, achieves $\mathcal{R}(T) = O(\sqrt{NT \ln T})$ with $O(N \log T)$ batch queries.*

*Proof.* The proof follows [Agrawal and Goyal, 2017, Theorem 1.2] and adapted to the batch setting. For each sub-optimal arm $a \neq 1$, in the analysis of the algorithm we use two thresholds $x_a$ and $y_a$ such that $\mu_a < x_a < y_a < \mu_1$. These parameters respectively control the events that the estimate $\hat{\mu}_a$ and sample $\theta_a$ are not too far away from the mean of arm $a$, namely, $\mu_a$. To remind the notation in Definition A.7, $E_a^\mu(t)$ represents the event $\{\hat{\mu}_a(B(t)) \leq x_a\}$ and $E_a^\theta(t)$ represents the event $\{\theta_a(t) \leq y_a\}$. The probability of playing each arm will be upper bounded based on whether or not the above events are satisfied.

Furthermore, the threshold $y_a$ is also used in the definition of $p_{a,t}$ (see Definition A.8) and Lemma A.10 to bound the probability of playing any suboptimal arm $a \neq 1$ at the current step $t$ by a linear function of $p_{a,t}$. Additionally, in Lemma A.13 we show an upper bound for the probability of selecting arm $a$ in terms of $x_a$ and $y_a$, i.e., $L_a(T) := O(\ln T/d(x_a, y_a))$.

For the problem-independent setting, we need to set $x_a = \mu_a + \Delta_a/3$ and $y_a = \mu_1 - \Delta_a/3$. This choice implies $\Delta_a'^2 = (\mu_1 - y_a)^2 = \Delta_a^2/9$. Then we can lower bound $d(x_a, \mu_a) \geq 2\Delta_a^2/9$. Thus $L_a(T) = O(\frac{\ln T}{\Delta_a^2})$. Now by substituting $\Delta_a$ and $d(x_a, \mu - a)$ in Theorem 4.3 for $a \neq 1$, we get $\mathbb{E}[k_a(T)] \leq O(\frac{\ln T}{\Delta_a^2})$. Now for arms with $\Delta_a > \sqrt{\frac{N \ln T}{T}}$, we can upper bound the regret by $\Delta_a \mathbb{E}[k_a(T)] = O(\sqrt{\frac{T \ln T}{N}})$, and for arms with $\Delta_a \leq \sqrt{\frac{N \ln T}{T}}$, we can upper bound the expected regret by $\sqrt{NT \ln T}$. All in all, it results in the total regret of $O(\sqrt{NT \ln T})$. $\square$

## A.4 Problem-independent Regret Bound with Gaussian Priors

**Theorem 4.7.** *Batch Thompson Sampling, outlined in Algorithm 1 and instantiated with Gaussian priors, achieves $\mathbb{E}[\mathcal{R}(T)] = O(\sqrt{NT \ln N})$ with $O(N \log T)$ batch queries.*

The proof is similar to the proof of Theorem 4.6 and follows essentially [Agrawal and Goyal, 2017, Theorem 1.3]. with Beta priors. We set $x_a = \mu_a + \Delta_a/3$ and $y_a = \mu_1 - \Delta_a/3$. The lemmas in the previous section for Beta priors hold here with slight modifications. The main lemma that changes for the Gaussian distributions is Lemma A.14.

**Lemma A.15.** *Let $\tau_j$ be the $j$-th time step in which the optimal arm 1 has been queried. Then*

$$\mathbb{E}\left[\frac{1}{p_{a,\tau_j+1}} - 1\right] \leq \begin{cases} e^{11} + 5, & \forall j, \\ \frac{4}{T\Delta_a^2}, & j > 8L_a(t), \end{cases}$$

*where $L_a(t) = \frac{18 \ln(T\Delta_a^2)}{\Delta_a^2}$.*

*Proof.* Note that $p_{a,t}$ is the probability $\Pr(\theta_a(t) > y_a | \mathcal{F}_{B(t)})$. If the prior comes from the Gaussian distribution then $\theta_a(t)$ has distribution $\mathcal{N}(\hat{\mu}_a(t), \frac{1}{k_a(B(t))+1})$. Given the definition of $\tau$ and $p_{a,t}$, the proof follows from Agrawal and Goyal [2017, Lemma 2.13]. $\square$

By using Lemma A.15 and substituting it in eq. (7), we can easily obtain the following lemma.

**Lemma A.16.** *For any arm $a \in [n]$ we have*

$$\sum_{t=1}^{T} \Pr(a(t) = a, E_a^\mu(t), E_a^\theta(t)) \leq (e^{64} + 4)(8L_a(t)) + \frac{8}{\Delta_a^2}.$$

**Lemma A.17.** *For any arm $a \in [n]$, we have*

$$\sum_{t=1}^{T} \Pr(a(t) = a, \overline{E_a^\mu(t)}) \leq \frac{1}{d(x_a, y_a)} + 1 \leq \frac{9}{2\Delta_a^2} + 1.$$

Similar to Lemma A.14, we can prove the following lemma.

**Lemma A.18.** *For any arm $a \in [n]$, we have*

$$\sum_{t=1}^{T} \Pr(a(t) = a, \overline{E_a^\theta(t)}, E_a^\mu(t)) \leq L_a(t) + \frac{1}{\Delta_a^2}.$$

*where* $L_a(t) = \frac{36\ln(T\Delta_a^2)}{\Delta_a^2}$.

*Proof.* The proof follows from [Agrawal and Goyal, 2017, Lemma 2.16] and is adapted to the batch setting. The decomposition is as in Lemma A.14. As before, the first term in the decomposition can be upper bounded by $L_a(t)$. Instead of bounding the second term with 1, we should bound it with $1/\Delta_a^2$. First, note that

$$\sum_{t=1}^{T} \Pr\left(a(t) = a, k_a(B(t)) > L_a(t), \overline{E_a^\theta(t)}, E_a^\mu(t)\right) \leq \mathbb{E}\left[\sum_{t=1}^{T} \Pr(\theta_a(t) > y_a | k_a(B(t)) > L_a(t), \hat{\mu}_a(B(t)) \leq x_a), \mathcal{F}_{B(t)}\right].$$

We also know that $\theta_a(t)$ is distributed as $\mathcal{N}(\hat{\mu}_a(t), \frac{1}{k_a(B(t))+1})$. So given $\{\hat{\mu}_a(t) \leq x_a\}$, we have that $\theta_a(t)$ is stochastically dominated by $\mathcal{N}(x_a, \frac{1}{k_a(B(t))+1})$. Therefore,

$$\Pr(\theta_a(B(t)) > y_a | k_a(B(t)) > L_a(t), \hat{\mu}_a(B(t)) \leq x_a, \mathcal{F}_{B(t)}) \leq \Pr\left(\mathcal{N}\left(x_a, \frac{1}{k_a(B(t))+1}\right) > y_a | \mathcal{F}_{B(t)}, k_a(B(t)) > L_a(t)\right).$$

By using concentration bounds, we have

$$\Pr\left(\mathcal{N}\left(x_a, \frac{1}{k_a(B(t))+1}\right) > y_a\right) \leq \frac{1}{2}e^{-\frac{L_a(t)(y_a - x_a)^2}{4}} \leq \frac{1}{T\Delta_a^2}.$$

Thus,

$$\Pr(\theta_a(t) > y_a | k_a(B(t)) > L_a(t), \hat{\mu}_a(t) \leq x_a, \mathcal{F}_{B(t)}) \leq 1/T\Delta_a^2. \qquad (9)$$

Summing over $t$ will follow the result. $\qquad \square$

Using lemmas A.18, A.16, A.17 we can upperbound

$$\mathbb{E}[k_a(t)] \leq (e^{64} + 4)\frac{2 \times 72\ln(T\Delta_a^2)}{\Delta_a^2} + \frac{2 \times 4}{\Delta_a^2} + \frac{18\ln(T\Delta_a^2)}{\Delta_a^2} + \frac{1}{\Delta_a^2} + \frac{9}{\Delta_a^2} + 1.$$

Thus, we can upper bound the expected regret due to arm $a$. Similar to the previous proofs we can upper bound

$$\Delta_a \mathbb{E}[k_i(T)] \leq O\left(\frac{1}{\Delta_a} + \frac{\ln(T\Delta_a^2)}{\Delta_a}\right) + \Delta_a.$$

Then, if $\Delta_a > e\sqrt{\frac{N\ln N}{T}}$ we can upper bound the regret by $O(\sqrt{\frac{N\ln T}{N}} + 1)$. If $\Delta_a \leq e\sqrt{\frac{N\ln N}{T}}$ we can upper bound the regret with $O(\sqrt{NT\ln T})$. Consequently, we can upper bound the total regret by $O(\sqrt{NT\ln T})$ assuming $T \geq N$.

## B  Batch Minimax Optimal Thompson Sampling

In order to increase clarity, we first introduce the main notations used in the proofs. We follow closely the notations used in Jin et al. [2020] and adapt them to the batch setting.

### B.1  Notations and Definitions

Without loss of generality, we assume the optimal arm is arm $a = 1$ with $\mu_1 = \max_{a \in [N]} \mu_a$.

**Definition B.1.** Define $\hat{\mu}_{as}$ to be the average reward of arm $a$ when it has been played $s$ times.

**Definition B.2.** We denote by $\mathcal{F}_s$ the history of plays of Algorithm 3 (B-MOTS) up to the $s$-th pull of arm 1.

**Definition B.3.** Let $h(j)$ be the largest power of 2 that is less than or equal to $j$.

**Definition B.4.** Define

$$\mathfrak{B} = \{s = 2^i | i = 0, \cdots, \log T\}.$$

We slightly modify Jin et al. [2020, eq.(16)] as follows.

**Definition B.5.** Define

$$\Delta = \mu_1 - \min_{s \in \mathfrak{B}} \left\{ \hat{\mu}_{1s} + \sqrt{\frac{\alpha}{s} \log^+ \left( \frac{T}{sN} \right)} \right\} . \tag{10}$$

**Definition B.6.** Similar to the definitions of $D_a(t)$ and $\theta_a(t)$, we define $D_{as}$ as the distribution of arm $a$ when it is played for the $s$-th time. Also, we define $\theta_{as}$ as a sample from distribution $D_{as}$.

**Lemma B.7.** *Let $X_1, X_2, \cdots$ be independent 1-subgaussian random variables with zero mean. Let's define $\hat{\mu}_t = 1/t \sum_{s=1}^{t} X_s$. Then for $\alpha \geq 4$ and any $\Delta > 0$*

$$\Pr \left( \exists s \in \mathfrak{B} : \hat{\mu}_s + \sqrt{\frac{\alpha}{s} \log^+(T/sN)} + \Delta \leq 0 \right) \leq \frac{15N}{T\Delta^2}.$$

The above lemma follows immediately from Lattimore and Szepesvári [2020, Lemma 9.3] as we consider $\mathfrak{B} \subseteq [T]$. We can strengthen Lemma B.7 for Gaussian variables, as described by Jin et al. [2020, Lemma 1] as follows.

**Lemma B.8.** *Let $X_a$'s be independent Guassian r.v. with zero mean and variance 1. Denote $\hat{\beta}_t = 1/t \sum_{s=1}^{t} X_s$. Then for $\alpha > 2$ and any $\Delta > 0$,*

$$\Pr \left( \exists s \in \mathfrak{B} : \hat{\beta}_s + \sqrt{\frac{\alpha}{s} \log^+(T/sN)} + \Delta \leq 0 \right) \leq \frac{4N}{T\Delta^2} .$$

Now similar to eq.(19) in Jin et al. [2020], define $\tau_{as}$ as follows.

**Definition B.9.** Define

$$\tau_{as} = \hat{\mu}_{as} + \sqrt{\frac{\alpha}{s} \log^+(\frac{T}{sN})} . \tag{11}$$

**Definition B.10.** We define $F_{as}$ as the CDF of distribution for arm $a$ when $k_a(t-1) = s$. Also $G_{as}(\epsilon)$ is defined as $1 - F_{as}(\mu_1 - \epsilon)$.

**Definition B.11.** Let us define $F'_{as}$ to be the $CDF$ of $\mathcal{N}(\hat{\mu}_{as}, 1/(\rho s))$. Moreover, let us define $G'_{as}(\epsilon) = 1 - F'_{as}(\mu_1 - \epsilon)$. Let $\tilde{\theta}_{as}$ denote a sample from $\mathcal{N}(\hat{\mu}_{as}, 1/(\rho s))$.

**Definition B.12.** Define the event $E_a(t) = \{\theta_a(t) \leq \mu_1 - \epsilon\}$.

The following two lemata deal with concentration inequalities that we need for subGaussian random variables.

**Lemma B.13** (Jin et al. [2020], Lemma 2). *Let $w > 0$ be a constant and $X_1, X_2, \cdots$ be independent and 1-subGaussian r.v. with zero mean. Denote by $\hat{\mu}_n = \frac{1}{n} \sum_{s=1}^{n} X_s$. Then for $\alpha > 0$ and any $N \leq T$,*

$$\sum_{n=1}^{T} \Pr \left( \hat{\mu}_n + \sqrt{\frac{\alpha}{n} \log^+(N/n)} \geq w \right) \leq 1 + \frac{\alpha \log^+(Nw^2)}{w^2} + \frac{3}{w^2} + \frac{\sqrt{2\alpha \log^+(Nw^2)}}{w^2}.$$

The following lemma is a variant of Jin et al. [2020, Lemma4].

**Lemma B.14.** *Let $\rho \in (1/2, 1)$ be a constant and $\epsilon > 0$. Assuming the reward of each arm is 1-sbuGaussian with mean $\mu_a$. For any fixed $\rho \in (1/2, 1)$ and $\alpha > 4$, there exists a constant $c > 0$ s.t.*

$$\mathbb{E} \left[ \sum_{s=1}^{T-1} \left( \frac{1}{G'_{1h(s)}(\epsilon)} - 1 \right) \right] \leq \frac{c}{\epsilon^2}. \tag{12}$$

*Proof.* The proof closely follows the steps of Jin et al. [2020, Lemma4]. However, for completeness, and for a few differences, we provide the full proof. The main difference is that in Lemma B.14 we have the terms $G'_{1h(s)}$ instead of $G'_{1s}$. We will prove the following two parts:

- First, there exists a constant $c'$ such that

$$\mathbb{E}\left[\frac{1}{G'_{1h(s)}(\epsilon)} - 1\right] \leq c', \ \forall s,$$

and

- Second, for $L = \lceil 64/\epsilon^2 \rceil$, we have

$$\mathbb{E}\left[\sum_{s=L}^{T}(\frac{1}{G'_{1h(s)}(\epsilon)} - 1)\right] \leq \frac{4}{e^2}(1 + \frac{16}{\epsilon^2}) \ .$$

Denote by $\Theta_s = \mathcal{N}(\hat{\mu}_{1h(s)}, 1/(\rho h(s)))$. Also, let $Y_s$ be the number of trials until a sample from $\Theta_s$ becomes greater than $\mu_1 - \epsilon$. By the definition of $G'_{ah(s)}$ we have

$$\mathbb{E}\left[\frac{1}{G'_{1h(s)}(\epsilon)} - 1\right] = \mathbb{E}\left[Y_s\right].$$

Similar to [Jin et al., 2020, Eq. (59)] one can show that

$$\Pr(Y_s < r) \geq 1 - r^{-2} - r^{-\frac{\rho'}{\rho}} \ .$$

Define $z = \sqrt{2\rho' \log r}$, for $r \geq 1$, where $\rho' \in (\rho, 1)$. Also let $M_r$ be the maximum of $r$ independent samples from $\Theta_s$. Thus

$$\Pr(Y_s < r) \geq \Pr(M_r > \mu_1 - \epsilon)$$

$$\geq \mathbb{E}\left[\mathbb{E}\left[\mathbb{I}(M_r > \hat{\mu}_{1h(s)} + \frac{z}{\sqrt{\rho h(s)}}, \hat{\mu}_{1h(s)} + \frac{z}{\sqrt{\rho h(s)}} \geq \mu_1 - \epsilon)\Big| \mathcal{F}_{h(s)}\right]\right]$$

$$= \mathbb{E}\left[\mathbb{I}(\hat{\mu}_{1h(s)} + \frac{z}{\sqrt{\rho h(s)}} \geq \mu_1 - \epsilon) \times \Pr\left(M_r > \hat{\mu}_{1h(s)} + \frac{z}{\sqrt{\rho h(s)}}\Big| \mathcal{F}_{h(s)}\right)\right] \ .$$

For a random variable $Z \sim \mathcal{N}(\mu, \sigma^2)$ we have the following tail bound

$$\Pr(Z > \mu + x\sigma) \geq \frac{1}{\sqrt{2\pi}}\frac{x}{x^2 + 1}e^{-\frac{x^2}{2}} \ .$$

Thus, for $r > e^2$,

$$\Pr\left(M_r > \hat{\mu}_{1h(s)} + \frac{z}{\sqrt{\rho h(s)}}\Big| \mathcal{F}_{h(s)}\right) \geq 1 - \exp\left(-\frac{r^{1-\rho'}}{\sqrt{8\pi \log r}}\right) \ .$$

Similar to Jin et al. [2020], we can show that if $r \geq \exp(10/(1 - \rho')^2)$ we have

$$\Pr\left(M_r > \hat{\mu}_{1h(s)} + \frac{z}{\sqrt{\rho h(s)}}\Big| \mathcal{F}_{h(s)}\right) \geq 1 - \frac{1}{r^2} \ .$$

Also, for $\epsilon > 0$, we have

$$\Pr\left(\hat{\mu}_{1h(s)} + \frac{z}{\sqrt{\rho h(s)}} \geq \mu_1 - \epsilon\right) \geq 1 - r^{-\rho'/\rho} \ .$$

Therefore, for $r \geq \exp(10/(1 - \rho')^2)$, we obtain

$$\Pr(Y_s < r) \geq 1 - r^{-2} - r^{-\rho'/\rho} \ .$$

For any $\rho' > \rho$ we get

$$\mathbb{E}\left[Y_s\right] = \sum_{r=0}^{\infty} \Pr(Y_s \geq r) \leq 2\exp\left(\frac{10}{(1 - \rho')^2}\right) + \frac{1}{(1 - \rho) - (1 - \rho')} \ .$$

By setting $1 - \rho' = (10\rho)/2$,

$$\mathbb{E}\left[\frac{1}{G'_{1h(s)}(\epsilon)} - 1\right] \leq 2\left(\frac{40}{(1-\rho)^2}\right) + \frac{2}{1-\rho} \ .$$

Now because $\rho$ is fixed, there exists a universal constant $c' > 0$ s.t.

$$\mathbb{E}\left[\frac{1}{G'_{1h(s)}(\epsilon)} - 1\right] \leq c' \ .$$

Proof of the second part is similar. $\hfill\square$

In the above proof, we had to be careful about the conditional expectations as the history in the batch mode, namely, $\mathcal{F}_{h(s)}$, is different from the sequential setting $\mathcal{F}_s$. Apart from that, as we stated, the proof is identical to Jin et al. [2020, Lemma4].

## B.2 Clipped Gaussian Distribution

**Theorem 5.1.** *If the reward of each arm is 1-subgussian then the regret of B-MOTS is bounded by* $\mathcal{R}(T) = O(\sqrt{NT} + \sum_{a:\Delta_a>0} \Delta_a)$. *Moreover, the number of batches is bounded by* $O(N\log T)$.

*Proof.* We closely follow the proof of of the fully sequential algorithm, provided in Jin et al. [2020, Theorem 1], and adapt it to the batch setting. Let us define

$$S := \{a : \Delta_a > \max\{2\Delta, 8\sqrt{N/T}\}\} \ .$$

Then, as Jin et al. [2020, eq. (17)] argued, we have

$$\mathcal{R}(T) \leq \sum_{a:\Delta_a>0} \Delta_a \mathbb{E}\left[k_a(t)\right]$$

$$\leq \mathbb{E}\left[2T\Delta\right] + 8\sqrt{NT} + \mathbb{E}\left[\sum_{a\in S} \Delta_a k_a(t)\right] \ . \tag{13}$$

where as in Jin et al. [2020, eq. (18)] (which immediately follows from Lemma B.8) we have $\mathbb{E}\left[2T\Delta\right] \leq 4/\sqrt{15NT}$. By Definition B.9, we have $\tau_{as} = \tau_a(t)$ when $k_a(t) = s$. Thus, for $a \in S$, we get

$$\tau_{1s} \geq \mu_1 - \Delta \geq \mu_1 - \frac{\Delta_a}{2}.$$

Therefore, for $\tilde{\theta}_{is}$ as defined in the definition B.11, we have

$$\Pr(\tilde{\theta}_{1s} \geq \mu_1 - \Delta_a/2) = \Pr(\theta_{1s} \geq \mu_1 - \Delta_a/2).$$

Hence for $a \in S$, we have

$$G_{1s}(\Delta_a/2) = G'_{1s}(\Delta_a/2).$$

For Algorithm 3, we need to revise Theorem 36.2 in Lattimore and Szepesvári [2020] as follows. Note that we start from $t = N + 1$ and $s = 1$ since the algorithm plays each arm once in the beginning.

**Lemma B.15.** *For $\epsilon > 0$, the expected number of times Algorithm 3 plays arm $a$ is bounded by*

$$\mathbb{E}\left[k_a(t)\right] \leq \mathbb{E}\left[\sum_{t=1}^{T} \mathbb{I}\{a(t) = a, E_a(t)\}\right] + \mathbb{E}\left[\sum_{t=1}^{T} \mathbb{I}\{a(t) = a, \overline{E_a(t)}\}\right]$$

$$\leq 1 + \mathbb{E}\left[\sum_{t=0}^{T-1}\left(\frac{1}{G_{1k_1(p_1(t))}} - 1\right)\mathbb{I}\{a(t) = 1\}\right] + \mathbb{E}\left[\sum_{t=N+1}^{T-1} \mathbb{I}\{a(t) = a, \overline{E_a(t)}\}\right] \tag{14}$$

$$\leq 2 + \mathbb{E}\left[\sum_{s=0}^{T-1}\left(\frac{1}{G_{1h(s)}(\epsilon)} - 1\right)\right] + \mathbb{E}\left[\sum_{s=0}^{T-1} \mathbb{I}\{G_{ah(s)}(\epsilon) > 1/T\}\right] \ . \tag{15}$$

*Proof.* We follow the steps in Lattimore and Szepesvári [2020] and make appropriate modifications for our batch mode algorithm. As defined in Definition B.12, $E_a(t) = \{\theta_a(t) \leq \mu_1 - \epsilon\}$. Thus,

$$\Pr(\theta_1(t) \geq \mu_1 - \epsilon | \mathcal{F}_{B(t)}) = G_{1k_1(B(t))} \ .$$

Now we consider the following decomposition based on $E_a(t)$ as follows,

$$\mathbb{E}\left[k_a(t)\right] = \mathbb{E}\left[\sum_{t=1}^{T} \mathbb{I}\{a(t) = a, E_a(t)\}\right] + \mathbb{E}\left[\sum_{t=1}^{T} \mathbb{I}\{a(t) = a, \overline{E_a(t)}\}\right] \ . \tag{16}$$

An upper bound for the first terms is as follows. Let $a'(t) = \mathrm{argmax}_{a \neq 1} \theta_a(t)$. Then,

$$\begin{aligned}
\Pr(a(t) = 1, E_a(t) | \mathcal{F}_{B(t)}) &\geq \Pr(a'(t) = a, E_a(t), \theta_1(t) \geq \mu_1 - \epsilon | \mathcal{F}_{B(t)}) \\
&= \Pr(\theta_1(t) \geq \mu_1 - \epsilon | \mathcal{F}_{B(t)}) \Pr(a'(t) = a, E_a(t) | \mathcal{F}_{B(t)}) \\
&\geq \frac{G_{1k_1(B(t))}}{1 - G_{1k_1(B(t))}} \Pr(a(t) = a, E_a(t) | \mathcal{F}_{B(t)}) \ .
\end{aligned}$$

In the first equality, we use the fact that $\theta_1(t)$ is conditionally independent of $a'(t)$ and $E_a(t)$, given $\mathcal{F}_{B(t)}$. For the second inequality we use

$$\Pr(a(t) = a, E_a(t) | \mathcal{F}_{B(t)}) \leq (1 - \Pr(\theta_1(t) > \mu_1 - \epsilon | \mathcal{F}_{B(t)})) \Pr(a'(t) = a, E_a(t) | \mathcal{F}_{B(t)}) \ .$$

Therefore,

$$\Pr(a(t) = a, E_a(t) | \mathcal{F}_{B(t)}) \leq \left(\frac{1}{G_{1k_1(B(t))}} - 1\right) \Pr(a(t) = 1 | \mathcal{F}_{B(t)}) \ .$$

By substituting this into (16), we obtain

$$\begin{aligned}
\mathbb{E}\left[\sum_{t=1}^{T} \mathbb{I}\{a(t) = a, E_a(t)\}\right] &\leq \mathbb{E}\left[\sum_{t=1}^{T} (\frac{1}{G_{1k_1(B(t))}} - 1)\mathbb{I}\{a(t) = 1\} | \mathcal{F}_{B(t)}\right] \\
&= \mathbb{E}\left[\sum_{t=1}^{T} (\frac{1}{G_{1k_1(B(t))}} - 1)\mathbb{I}(a(t) = 1)\right] \\
&\leq \mathbb{E}\left[\sum_{s=0}^{T-1} (\frac{1}{G_{1h(s)}} - 1)\right] \ .
\end{aligned}$$

Now define

$$\tau = \{t \in [T] : 1 - F_{ak_a(B(t))}(\mu_1 - \epsilon) > 1/T\} \ .$$

For the second expression in (16), we get

$$\begin{aligned}
\mathbb{E}\left[\sum_{t=1}^{T} \mathbb{I}(a(t) = a, \overline{E_a(t)})\right] &\leq \mathbb{E}\left[\sum_{t \in \tau} \mathbb{I}(a(t) = a)\right] + \mathbb{E}\left[\sum_{t \notin \tau} \mathbb{I}(\overline{E_a(t)})\right] \\
&\leq \mathbb{E}\left[\sum_{s=0}^{T-1} \mathbb{I}\{1 - F_{ah(s)}(\mu_1 - \epsilon)\} > 1/T\right] + \mathbb{E}\left[\sum_{t \notin \tau} \frac{1}{T}\right] \\
&\leq \mathbb{E}\left[\sum_{s=0}^{T-1} \mathbb{I}(G_{ah(s)} > 1/T)\right] + 1.
\end{aligned}$$

$\square$

Now by setting $\epsilon = \Delta_a/2$ we can show that

$$\Delta_a \mathbb{E}\left[k_a(t)\right] \leq \Delta_a + \Delta_a \mathbb{E}\left[\sum_{N+1}^{T-1} \mathbb{I}\{a(t) = a, \overline{E_a(t)}\}\right] + \Delta_a \mathbb{E}\left[\sum_{t=1}^{T-1} (\frac{1}{G'_{1k_1(B(t))}(\Delta_a/2)} - 1)\mathbb{I}(a(t) = 1)\right]. \tag{17}$$

To bound the first term we note that

$$\overline{E_a(t)} \subseteq \left\{ \hat{\mu}_a(B(t)) + \sqrt{\frac{\alpha}{k_a(B(t))} \log^+ \left( \frac{T}{Nk_a(B(t))} \right)} > \mu_1 - \Delta_a/2 \right\} .$$

Define $\kappa_a$ as the sum of the event in the right hand side of the above equation, namely,

$$\kappa_a = \sum_{s=1}^{T} \mathbb{I} \left\{ \hat{\mu}_{ah(s)} + \sqrt{\alpha/h(s) \log^+(T/h(s)N)} > \mu_1 - \frac{\Delta_a}{2} \right\} . \tag{18}$$

Hence,

$$\Delta_a \mathbb{E} \left[ \sum_{N+1}^{T-1} \mathbb{I}\{a(t) = a, \overline{E_a(t)}\} \right] \leq \Delta_a \mathbb{E}\left[\kappa_a\right] = \Delta_a \mathbb{E} \left[ \sum_{s=1}^{T} \mathbb{I} \left\{ \hat{\mu}_{ah(s)} + \sqrt{\frac{\alpha}{h(s)} \log^+ \left( \frac{T}{h(s)N} \right)} > \mu_1 - \Delta_a/2 \right\} \right] .$$

Using

Lemma B.13 and the fact that $\Delta_a = \mu_1 - \mu_a$ we have

$$\Delta_a \mathbb{E}\left[\kappa_a\right] \leq \Delta_a \sum_{s=1}^{T} \Pr \left\{ \hat{\mu}_{ah(s)} - \mu_a + \sqrt{\frac{\alpha}{h(s)} \log^+(T/h(s)N)} > \frac{\Delta_a}{2} \right\} \tag{19}$$

$$\leq \Delta_a + \frac{12}{\Delta_a} + \frac{4\alpha}{\Delta_a} \left( \log^+(\frac{T\Delta_a^2}{4N}) + \sqrt{2\alpha\pi \log^+(\frac{T\Delta_a^2}{4N})} \right) . \tag{20}$$

Now it implies that $\mathbb{E}\left[\Delta_a \kappa_a\right] = O(\sqrt{T/k} + \Delta_a)$. For bounding the second term of (17), a slight modification of Lemma B.14, provides

$$\Delta_a \mathbb{E} \left[ \sum_{t=1}^{T-1} \left( \frac{1}{G'_{1k_1(B(t))}(\Delta_a/2)} - 1 \right) \mathbb{I}(a(t) = 1) \right] = \Delta_a \mathbb{E} \left[ \sum_{s=1}^{T-1} \left( \frac{1}{G'_{1h(s)}(\Delta_a/2)} - 1 \right) \right] = O(\sqrt{T/N}) .$$

$\square$

### B.3 MOTS 1-subgaussian asymptotic regret bound

**Theorem 5.2.** *Assume that the reward of each arm $a \in [N]$ is 1-subgaussian with mean $\mu_a$. For any fixed $\rho \in (1/2, 1)$, the regret of B-MOTS can be bounded as $\mathcal{R}(T) = O \left( \log(T) \sum_{a:\Delta_a > 0} \frac{1}{\rho \Delta_a} \right)$.*

First we should prove the following lemma, which a simple variant of Jin et al. [2020, Lemma 6] for the batch setting.

**Lemma B.16.** *For any $\epsilon_T > 0$, and $\epsilon > 0$ that satisfies $\epsilon + \epsilon_T < \Delta_a$, it holds that*

$$\mathbb{E} \left[ \sum_{s=1}^{T-1} \mathbb{I}\{G'_{ah(s)} > 1/T\} \right] \leq 1 + \frac{4}{\epsilon_T^2} + \frac{4 \log T}{\rho(\Delta_a - \epsilon - \epsilon_T)^2} .$$

*Proof.* The proof closely follows Jin et al. [2020, Lemma 6] and adapted to the batch setting. As before $\mu_a + \epsilon_T \leq \mu_1 - \epsilon$, and by using the tail-bound for $\sigma$-subGaussian random variables we have

$$\Pr(\hat{\mu}_{ah(s)} > \mu_a + \epsilon_T) \leq \exp(-h(s)\epsilon_T^2/2) \leq \exp(-s\epsilon_T^2/4).$$

Furthermore

$$\sum_{s=1}^{\infty} \exp \left( -\frac{s\epsilon_T^2}{4} \right) \leq 4/\epsilon_T^2.$$

Define

$$L_a = 4 \log T/(\rho(\Delta_a - \epsilon - \epsilon_T)^2).$$

For $s \geq L_a$, let $X_{as}$ be sampled from $\mathcal{N}(\hat{\mu}_{ah(s)}, 1/(\rho h(s)))$. Then if we have $\hat{\mu}_{ah(s)} \leq \mu_a + \epsilon_T$, the Guassian tail bound implies

$$\Pr(X_{as} \geq \mu_1 - \epsilon) \leq \frac{1}{2} \exp\left(-\frac{\rho h(s)(\Delta_a - \epsilon - \epsilon_T)^2}{2}\right) \leq 1/T.$$

Now, denote the event $\{\hat{\mu}_{ah(s)} \leq \mu_a + \epsilon_T\}$ by $Y_{as}$. By using the fact that $\Pr(A) \leq \Pr(A|B) + 1 - \Pr(B)$, we have

$$\mathbb{E}\left[\sum_{s=1}^{T-1} \mathbb{I}\{G'_{ah(s)}(\epsilon) > 1/T\}\right] = \sum_{s=1}^{T-1} \Pr(\{G'_{ah(s)}(\epsilon) > 1/T\})$$

$$\leq \sum_{s=1}^{T-1} \Pr(\{G'_{ah(s)}(\epsilon) > 1/T\}|Y_{as}) + \sum_{s=1}^{T-1}(1 - \Pr(Y_{as}))$$

$$\leq \lceil L_a \rceil + \sum_{s=1}^{T-1}(1 - \Pr(Y_{as}))$$

$$\leq 1 + \frac{4}{\epsilon_T^2} + \frac{4 \log T}{\rho(\Delta_a - \epsilon - \epsilon_T)^2} .$$

$\square$

Now, closely following the proof of Jin et al. [2020, Theorem 2], we define

$$Z(\epsilon) = \left\{\forall s \in \mathfrak{B} : \hat{\mu}_{1s} + \sqrt{\frac{\alpha}{s} \log^+(\frac{T}{sN})} \geq \mu_1 - \epsilon\right\} . \tag{21}$$

For an arm $a \in [N]$, we have

$\mathbb{E}[k_a(t)] \leq \mathbb{E}[k_a(t)|Z(\epsilon)] \Pr(Z(\epsilon)) + T(1 - \Pr(Z(\epsilon)))$

$$\leq 2 + \mathbb{E}\left[\sum_{s=1}^{T-1}(\frac{1}{G_{1h(s)}(\epsilon)} - 1)|Z(\epsilon)\right] + T(1 - \Pr(Z(\epsilon))) + \mathbb{E}\left[\sum_{s=1}^{T-1}\mathbb{I}(G_{ah(s)}(\epsilon) > 1/T)\right]$$

$$\leq 2 + \mathbb{E}\left[\sum_{s=1}^{T-1}(\frac{1}{G'_{1h(s)}(\epsilon)})\right] + T(1 - \Pr(Z(\epsilon))) + \mathbb{E}\left[\sum_{s=1}^{T-1}\mathbb{I}(G'_{ah(s)}(\epsilon) > 1/T)\right] .$$

The second inequality is due to Lemma B.15 and the last inequality is due to the fact that given $Z(\epsilon)$, we have $G_{1h(s)}(\epsilon) = G'_{1h(s)}(\epsilon)$. Also, note that if

$$\hat{\mu}_{ah(s)} + \sqrt{\frac{\alpha}{h(s)} \log^+(T/h(s)N)} \geq \mu_1 - \epsilon,$$

then we have $G_{ah(s)}(\epsilon) = G'_{ah(s)}(\epsilon)$, or otherwise we have $G_{ah(s)}(\epsilon) = 0 \leq G'_{as}(\epsilon)$.

Now from Lemma B.7 and by setting $\epsilon = \epsilon_T = \frac{1}{\log \log T}$, we have

$$T(1 - \Pr(Z(\epsilon))) \leq 15N(\log \log T)^2.$$

By using Lemma B.14

$$\mathbb{E}\left[\sum_{s=1}^{T-1}(\frac{1}{G'_{1h(s)}(\epsilon)} - 1)\right] \leq O((\log \log T)^2) .$$

Then, by Lemma B.16

$$\mathbb{E}\left[\sum_{s=1}^{T-1}\mathbb{I}(G'_{ah(s)}(\epsilon) > 1/T)\right] \leq 1 + 4(\log \log T)^2 + \frac{4 \log T}{\rho(\Delta_a - 2/\log \log T)^2} .$$

The theorem will follow easily by combining the above equations, namely,

$$\lim_{T \to \infty} \frac{\mathbb{E}[\Delta_a k_a(t)]}{\log T} = \frac{2}{\rho \Delta_a} .$$

## B.4 MOTS for Gaussian Rewards

**Theorem 5.3.** *Assume that the reward of each arm $a$ is sampled from a Gaussian distribution $\mathcal{N}(\mu_a, 1)$ and $\alpha > 2$. Then, the regret of B-MOTS-J can be bounded as follows:*

$$\mathcal{R}(T) = O(\sqrt{KT} + \sum_{a=2}^{k} \Delta_a), \quad \lim_{T\to\infty} \frac{\mathcal{R}(T)}{\log(T)} = \sum_{a:\Delta_a>0} \frac{2}{\Delta_a}.$$

Recall that $F'_{as}$ denotes the CDF of $\mathcal{J}(\hat{\mu}_{as}, 1/s)$ for any $s \geq 1$ and $G'_{as} = 1 - F'_{as}(\mu_1 - \epsilon)$. We closely follow the recipe of [Jin et al., 2020, Theorem 4]. The proof of the minimax and asymptotic-optimal bounds are similar to the proof of Theorem 5.1 and 5.2 with a few differences. Note that in the proof of Theorem 5.1, we used the fact that $\rho < 1$ (used in the definition of the Gaussian distribution $\tilde{\theta}_a$). In Theorem 5.3, we do not have the parameter $\rho$. Therefore instead of Lemma B.14 we prove the following, which is a batch variant of Jin et al. [2020, Lemma 9].

**Lemma B.17.** *There exists a universal constant $c$, s.t.,*

$$\mathbb{E}\left[\sum_{s=1}^{T-1} \left(\frac{1}{G'_{1h(s)}(\epsilon)} - 1\right)\right] \leq c/\epsilon^2 .$$

*Proof.* Similar to (Lemma B.14), the following two statements need to be proven:
(i) there exists a universal constant $c'$ s.t.

$$\sum_{s=1}^{L} \mathbb{E}\left[\frac{1}{G'_{1h(s)}(\epsilon)} - 1\right] \leq \frac{c'}{\epsilon^2}, \forall s .$$

(ii) for $L = \lceil 64/\epsilon^2 \rceil$

$$\mathbb{E}\left[\sum_{s=L}^{T} \left(\frac{1}{G'_{1h(s)}(\epsilon)} - 1\right)\right] \leq \frac{4}{\epsilon^2}(1 + 16/\epsilon^2) .$$

The proof of statement (ii) is similar to the one in Lemma B.8. Therefore, We focus on the first statement here, which closely follows the proof of Jin et al. [2020, Lemma 9].

Let $\hat{\mu}_{1h(s)} = \mu_1 + x$. Let $Z$ be a sample from $\mathcal{J}(\hat{\mu}_{1h(s)}, 1/h(s))$. For $x < -\epsilon$, applying Lemma B.8 with $z = -\sqrt{h(s)}(\epsilon + x) > 0$ we have

$$G'_{1h(s)}(\epsilon) = \Pr(Z > \mu_1 - \epsilon) = \frac{1}{2}\exp\left(-\frac{h(s)(\epsilon+x)^2}{2}\right) . \tag{22}$$

Note that $x \sim \mathcal{N}(0, 1/h(s))$. Let $f(x)$ be the PDF of $\mathcal{N}(0, 1/h(s))$.

$$\mathbb{E}_{x\sim\mathcal{N}(0,1/h(s))}\left[\left(\frac{1}{G'_{1h(s)}(\epsilon)} - 1\right)\right] = \int_{\infty}^{-\epsilon} f(x)\left(\frac{1}{G'_{1h(s)}(\epsilon)} - 1\right)dx + \int_{-\epsilon}^{-\infty} f(x)\left(\frac{1}{G'_{1h(s)}(\epsilon)} - 1\right)dx$$

$$\leq \int_{-\infty}^{-\epsilon} f(x)\left(2\exp\left(\frac{h(s)(\epsilon+x)^2}{2}\right) - 1\right)dx + \int_{-\epsilon}^{\infty} f(x)\left(\frac{1}{G'_{1h(s)}}(\epsilon) - 1\right)dx$$

$$\leq \int_{-\infty}^{-\epsilon} f(x)\left(2\exp\left(\frac{h(s)(\epsilon+x)^2}{2}\right) - 1\right)dx + \int_{-\epsilon}^{\infty} f(x)dx$$

$$\leq \sqrt{2}\frac{e^{-s\epsilon^2/4}}{\sqrt{s\epsilon}} + 1$$

The first inequality is because of eq. (22). The second inequality is because $G'_{1h(s)}(\epsilon) = \Pr(Z > \mu_1 - \epsilon) \geq 1/2$, since $\hat{\mu}_{1h(s)} = \mu_1 + x \geq \mu_1 - \epsilon$. And the last inequality is due to the definition of $h(s)$.

Also for $s \leq L$, we have $e^{-s\epsilon^2/4} = O(1)$, thus for $L = \left\lceil \frac{64}{\epsilon^2} \right\rceil$,

$$\sum_{s=1}^{L} \mathbb{E}\left[\left(\frac{1}{G'_{1h(s)}(\epsilon)} - 1\right)\right] = O\left(\sum_{s=1}^{L} \frac{1}{\sqrt{s}\epsilon}\right) = O(1/\epsilon^2).$$

$\square$

From the above lemma we have

$$\Delta_a \mathbb{E}\left[\sum_{s=1}^{T-1}\left(\frac{1}{G'_{1h(s)}}(\Delta_a/2) - 1\right)\right] \leq O(\sqrt{T/K} + \Delta_a).$$

The rest of the proof for minimax optimality is similar to the proof of Theorem 5.1.

For the asymptotic regret bound, we first state the following lemma, which the batch mode version of

**Lemma B.18.** *for any $\epsilon_T > 0, \epsilon > 0$ that satisfies $\epsilon + \epsilon_T < \Delta_a$, we have*

$$\mathbb{E}\left[\sum_{s=1}^{T-1} \mathbb{I}\{G'_{ih(s)} > 1/T\}\right] \leq 1 + \frac{4}{\epsilon_T^2} + \frac{4\log T}{(\Delta_a - \epsilon - \epsilon_T)^2}.$$

*Proof.* The proof is similar to the proof of Lemma B.16. $\square$

The proof asymptotic regret bound is similar to the proof of Theorem 5.2 where we use Lemmas B.17, B.8, and B.18.

## C   Batch Thompson Sampling for Contextual Bandits

First, we reintroduce a number of notations from Agrawal and Goyal [2013b] and adapt them to the batch setting.

### C.1   Notations and Definitions

In time step $t$ of the B-TS-C algorithm, we generate a sample $\tilde{\mu}(t)$ from $\mathcal{N}(\hat{\mu}(B(t)), v^2\mathcal{B}(B(t))^{-1})$ and play the arm $a$ with maximum $\theta_a(t) = b_a(t)^T \tilde{\mu}(t)$.

**Definition C.1.** Let us define the standard deviation of empirical mean in the batch setting as

$$s_a(B(t)) := \sqrt{b_a(t)^T \mathcal{B}(B(t))^{-1} b_a(t)}.$$

**Definition C.2.** Let us define the history of the process up to time $t$ by

$$H_t = \{a(\tau), r_{a(\tau)}(\tau), b_a(\tau) | a \in [N], \tau \in [t]\},$$

where $a(\tau)$ indicates the arm played at time $\tau$, $b_a(\tau)$ indicates the context vector associated with arm $a$ at time $\tau$, and $r_{a(\tau)}$ indicates the reward at time $\tau$.

**Definition C.3.** Define the filtration $\mathcal{F}_{B(t)}$ as the union of history until time $B(t)$, and the context vectors up to time $t$, i.e.,

$$\mathcal{F}_{B(t)} = \{H_{B(t)}, b_a(t') | a \in [N], t' \in (B(t), t]\}.$$

**Definition C.4.** We assume that $\eta_{a,t} = r_a(t) - \langle b_a(t), \mu \rangle$, conditioned on $\mathcal{F}_{B(t)}$, is $\sigma$-subGaussian for some $\sigma \geq 0$.

**Definition C.5.** Define

$$l(t) = \sigma\sqrt{d\ln\frac{t^3}{\delta}} + 1,$$

$$v(t) = \sigma\sqrt{9d\ln\frac{t}{\delta}},$$

$$p = \frac{1}{4e\sqrt{\pi}},$$

$$g(t) = \min\{\sqrt{4d\ln(t)}, \sqrt{4\log(tN)}\}v(t) + l(t).$$

**Definition C.6.** Define $E^\mu(t)$ as the event that for any arm $a$

$$\left\{|\langle b_a(t), \hat{\mu}(B(t)) - b_a(t)^\top \mu\rangle| \leq l(t)s_a(B(t))\right\}.$$

**Definition C.7.** Define $E^\theta(t)$ as the event

$$\left\{\forall a : |\theta_a(t) - \langle b_a(t), \hat{\mu}(B(t))\rangle| \leq (g(t) - l(t))s_a(B(t))\right\}.$$

**Definition C.8.** Define the difference between the mean reward of the optimal arm at time $t$, denoted by $a^*(t)$, and arm $a$ as follows

$$\Delta_a(t) = \langle b_{a^*(t)}(t), \mu\rangle - \langle b_a(t), \mu\rangle.$$

**Definition C.9.** We say that an arm is saturated at time $t$ if $\Delta_a(t) > g(t)s_a(B(t))$. We also denote by $C(t)$ the set of saturated arms at time $t$. An arm $a$ is unsaturated at time $t$ of $a \notin C(t)$.

**Lemma C.10** (Abbasi-Yadkori et al. [2011]). *Let $\mathcal{F}'_t$ be a filteration. Consider two random processes $m_t \in \mathbb{R}^d$ and $\mu_t \in \mathbb{R}$ where $m_t$ is $\mathcal{F}'_{t-1}$-measurebale and $\mu_t$ is a martingale difference process and $\mathcal{F}'_t$-measurebale. Define, $\xi_t = \sum_{\tau=1}^t m_\tau \mu_t$ and $M_t = I_d + \sum_{\tau=1}^t m_\tau m_\tau^\top$. Assume that given $\mathcal{F}'_t$, $\mu_t$ is $\sigma$-subGaussian. Then, with probability $1 - \delta$,*

$$\|\xi_t\|_{M_t^{-1}} \leq \sigma\sqrt{d\ln\frac{t+1}{\delta}}.$$

## C.2 Analysis

**Theorem 6.1.** *The B-TS-C algorithm (Algorithm 2) achieves the total regret of*

$$\mathcal{R}(T) = O\left(d^{3/2}\sqrt{T}(\ln(T) + \sqrt{\ln(T)\ln(1/\delta)})\right)$$

*with probability $1 - \delta$. Moreover, B-TS-C carries out $O(N\log T)$ batch queries.*

The proof closely follows [Agrawal and Goyal, 2013b, Theorem 1]. We first start with the following lemma, that is a batch version of [Agrawal and Goyal, 2013b, Lemma 1].

**Lemma C.11.** *For all $t$, and $0 < \delta < 1$, we have $\Pr(E^\mu(B(t))) \geq 1 - \delta/t^2$. Moreover, For all filtration $\mathcal{F}_{B(t)}$, we have $\Pr(E^\theta(t)|\mathcal{F}_{B(t)}) \geq 1 - 1/t^2$.*

*Proof.* The proof closely follows Agrawal and Goyal [2013b, Lemma 1] where we adapt it to the batch setting. We only prove the first part as the second part very similar. We first invoke Lemma C.10 as follows. Set $m_t = b_{a(t)}(t)$, $\eta_t = r_{a(t)}(t) - b_{a(t)}(t)^T\mu$, and

$$F'_t = \{a(\tau + 1), m_{\tau+1} : \tau \leq t\} \cup \{\eta_\tau : \tau \leq B(t)\}.$$

Note that $\eta_t$ is conditionally $\sigma$-subgaussian, and is a martingale difference process. Therefore,

$$\mathbb{E}\left[\eta_t|F'_{B(t)}\right] = \mathbb{E}\left[r_{a(t)}|b_{a(t)}(t), a(t)\right] - \langle b_{a(t)}(t), \mu\rangle = 0.$$

Thus, we have

$$M_t = I_d + \sum_{\tau=1}^t m_\tau m_\tau^\top$$

and

$$\xi_t = \sum_{\tau=1}^{t} m_\tau \eta_\tau .$$

Similar to Agrawal and Goyal [2013b, Lemma 1], we have $\mathcal{B}(t) = M_{t-1}$, but we need to change $\hat{\mu}(t) - \mu = M_{B(t)}^{-1}(\xi_{B(t)} - \mu)$. For any vector $y \in \mathbb{R}$ and matrix $A \in \mathbb{R}^{d \times d}$, let us define the norm $\|y\|_A := \sqrt{y^T A y}$. Hence, for all $a$,

$$|\langle b_a(t), \hat{\mu}(t) \rangle - \langle b_a(t), \mu \rangle| = \|b_a(t)\|_{\mathcal{B}(t)^{-1}} \times \|\xi_{B(t)} - \mu\|_{M_{B(t)}^{-1}} .$$

Since $B(t) \leq t - 1$, Lemma C.10 implies that with probability at least $1 - \delta'$,

$$\|\xi_{B(t)}\|_{M_{B(t)}^{-1}} \leq \sigma \sqrt{d \ln(t/\delta')}.$$

Thus,

$$\|\xi_{B(t)} - \mu\|_{M_{B(t)}^{-1}} \leq \sigma \sqrt{d \ln(t/\delta')} + \|\mu\|_{M_{B(t)}^{-1}} \leq \sigma \sqrt{d \ln(t/\delta')} + 1.$$

Now by setting $\delta' = \frac{\delta}{t^2}$ we have with probability $1 - \delta/t^2$, and for all arms $a$,

$$|\langle b_a(t), \hat{\mu}(B(t)) \rangle - \langle b_a(t), \mu \rangle| \leq l(t) s_a(B(t)) .$$

$\square$

Now, we lower bound the probability that $\theta_{a^*(t)}(t)$ becomes larger than $\langle b_{a^*(t)}(t), \mu \rangle$.

**Lemma C.12.** *For any filtration $\mathcal{F}_{B(t)}$, if $E^\mu(t)$ holds true, we have*

$$\Pr \left( \theta_{a^*(t)}(t) > \langle b_{a^*(t)}(t), \mu \rangle | \mathcal{F}_{B(t)} \right) \geq p.$$

*Proof.* The proof easily follows from Agrawal and Goyal [2013b, Lemma 2]. Suppose $E^\mu(t)$ holds true, then

$$|\langle b_{a^*(t)}(t), \hat{\mu}(t) \rangle - \langle b_{a^*(t)}(t), \mu \rangle| \leq \ell(t) s_{a^*(t)}(B(t)) .$$

The Gaussian random variable $\theta_{a^*(t)}(t)$ has mean $\langle b_{a^*(t)}(t), \hat{\mu}(t) \rangle$ and standard deviation $v_t s_{a^*(t)}(B(t))$. Therefore, we have

$$\Pr(\theta_{a^*(t)}(t) \geq \langle b_{a^*(t)}(t), \mu \rangle | \mathcal{F}_{B(t)}) \geq \frac{1}{4\sqrt{\pi}} e^{-Z_t^2} .$$

where $|Z_t| = \left| \frac{\langle b_{a^*(t)}(t), \hat{\mu}(t) \rangle - \langle b_{a^*(t)}(t), \mu \rangle}{v(t) s_{a^*(t)}(B(t))} \right| \leq 1$. $\square$

The following lemma bounds the probability that an arm played at time $t$ is not saturated.

**Lemma C.13.** *Given $\mathcal{F}_{B(t)}$, if $E^\mu(t)$ is true,*

$$\Pr(a(t) \notin C(t) | \mathcal{F}_{B(t)}) \geq p - \frac{1}{t^2} .$$

*Proof.* The proof is a slight modification of Agrawal and Goyal [2013b, Lemma 3] for the batch setting. If $\forall j \in C(t)$ we have $\theta_{a^*(t)}(t) > \theta_j(t)$, then one of the unsaturated actions much be played which leads us to

$$\Pr(a(t) \notin C(t) | \mathcal{F}_{B(t)}) \geq \Pr(\theta_{a^*(t)}(t) > \theta_j(t), \forall j \in C(t) | \mathcal{F}_{B(t)}).$$

Note that for all saturated arms $j \in C(t)$, we have

$$\Delta_j(t) > g(t) s_j(B(t)).$$

In the case that $E^\mu(t)$ and $E^\theta(t)$ are both true, we have

$$\theta_j(t) \leq \langle b_j(t), \mu \rangle + g(t) s_j(B(t)).$$

Hence, conditioned on $\mathcal{F}_{B(t)}$ if $E^\mu(t)$ is true, we have either the event $E^\theta(t)$ is false or for all $j \in C(t)$,

$$\theta_j(t) \leq \langle b_j(t), \mu \rangle + g(t) s_j(B(t)) \leq \langle b_{a^*(t)}(t), \mu \rangle,$$

Thus, for any $\mathcal{F}_{B(t)}$ that $E^\mu(t)$ holds,

$$\Pr(\theta_{a^*(t)}(t) > \theta_j(t), \forall j \in C(t)|\mathcal{F}_{B(t)}) \geq \Pr(\theta_{a^*(t)}(t) > \langle b_{a^*(t)}(t), \mu\rangle|\mathcal{F}_{B(t)}) - \Pr(\overline{E^\theta(t)}|\mathcal{F}_{B(t)})$$

$$\geq p - \frac{1}{t^2} .$$

The above inequalities are due to Lemmas C.11 and C.12. $\qquad\square$

**Lemma C.14.** *For any filtration $\mathcal{F}_{B(t)}$, assuming $E^\mu(t)$ holds true,*

$$\mathbb{E}\left[\Delta_{a(t)}(t)|\mathcal{F}_{B(t)}\right] \leq \frac{3g(t)}{p}\mathbb{E}\left[s_{a(t)}(B(t))|\mathcal{F}_{B(t)}\right] + \frac{2g(t)}{pt^2} .$$

*Proof.* The proof follows closely Agrawal and Goyal [2013b, Lemma 4] and adapts it to the batch setting. First define

$$\bar{a}(t) = \arg\min_{a \notin C(t)} s_a(B(t)),$$

Since $\mathcal{F}_{B(t)}$ defines $\mathcal{B}(B(t))$ and also $b_a(t)$ are independent of unobserved rewards (before making a batch query) thus given $\mathcal{F}_{B(t)}$ and context vectors $b_a(t)$, the value of $\bar{a}(t)$ is determined. Now by applying Lemma C.13, for any $\mathcal{F}_{B(t)}$ and by assuming that $E^\mu(\theta)$ is true, we have

$$\mathbb{E}\left[s_{a(t)}(B(t))|\mathcal{F}_{B(t)}\right] \geq \mathbb{E}\left[s_{a(t)}(B(t))|\mathcal{F}_{B(t)}, a(t) \notin C(t)\right] \cdot \Pr(a(t) \notin C(t)|\mathcal{F}_{T-1})$$

$$\geq s_{\bar{a}(t)}(B(t))(p - \frac{1}{t^2}).$$

Again if both $E^\mu(t)$ and $E^\theta(t)$ are true, then for all $a$ we have,

$$\theta_a(t) \leq \langle b_a(t), \mu\rangle + g(t)s_a(B(t)).$$

Moreover, we know that for all $a$, $\theta_{a(t)}(t) \geq \theta_a(t)$, thus

$$\Delta_{a(t)}(t) = \Delta_{\bar{a}(t)}(t) + (\langle b_{\bar{a}(t)}(t), \mu\rangle - \langle b_{a(t)}(t), \mu\rangle)$$

$$\leq 2g(t)s_{\bar{a}(t)}(B(t)) + g(t)s_{a(t)}(B(t)).$$

Consequently,

$$\mathbb{E}\left[\Delta_{a(t)}|\mathcal{F}_{B(t)}\right] \leq \frac{2g(t)}{p - \frac{4}{t^2}}\mathbb{E}\left[s_{a(t)}(B(t))|\mathcal{F}_{B(t)}\right] + g(t)\mathbb{E}\left[s_{a(t)}(B(t))|\mathcal{F}_{B(t)}\right] + \frac{1}{t^2}$$

$$\leq \frac{3}{p}g(t)\mathbb{E}\left[s_{a(t)}(B(t))|\mathcal{F}_{B(t)}\right] + \frac{2g(t)}{pt^2} .$$

The first inequality is because $\Delta_a \leq 1$ for all $a$. The second inequality uses Lemma C.11 to get $\Pr(\overline{E^\theta(t)}) \leq \frac{1}{t^2}$. Furthermore, in the last inequality we use the fact that $0 \leq s_{a(t)}(B(t)) \leq |b_{a(t)}(t)| \leq 1$. $\qquad\square$

Similar to Agrawal and Goyal [2017] we have the following definitions.

**Definition C.15.**
$$\mathcal{R}'(t) := \mathcal{R}(t) \times \mathbb{I}(E^\mu(t)).$$

**Definition C.16.** Define

$$X_t = \mathcal{R}'(t) - \frac{3g(t)}{p}s_{a(t)}(B(t)) - \frac{2g(t)^2}{pt^2}$$

$$Y_t = \sum_{w=1}^t X_w.$$

Becasue of the way we defined $Y_t$, namely, the filteration $\mathcal{F}_{B(t)}$, we can easily show the following lemma.

**Lemma C.17.** *The sequence $\{Y_t\}_{t=0}^T$ is a super martingale with respect to $\mathcal{F}_{B(t)}$.*

*Proof.* The proof follows closely Agrawal and Goyal [2013b, Lemma 5] and adapts it to filtration $\mathcal{F}_{B(t)}$ induced by the batch algorithm. Basically, we need to show that for all $t \geq 0$,

$$\mathbb{E}\left[Y_t - Y_{t-1} | \mathcal{F}_{B(t)}\right] \leq 0,$$

In other words

$$\mathbb{E}\left[\mathcal{R}'(t) | \mathcal{F}_{B(t)}\right] \leq \frac{3g(t)}{p}\mathbb{E}\left[s_{a(B(t))}(t) | \mathcal{F}_{B(t)}\right] + \frac{2g(t)}{pt^2},$$

First, note that $\mathcal{F}_{B(t)}$ determines the event $E^\mu(t)$. Assuming that $\mathcal{F}_{B(t)}$ is such that $E^\mu(t)$ is not true, then $\mathcal{R}'(t) = 0$ ] and the above inequality is trivial. Otherwise, if for $\mathcal{F}_{B(t)}$, the event $E^\mu(t)$ holds, Lemma C.14 implies the result. $\quad\square$

The following Lemma is a batch variant of Chu et al. [2011, Lemma 3].

**Lemma C.18.**

$$\sum_{t=1}^{T} s_{a(t)}(B(t)) \leq 5\sqrt{dT \ln T} . \tag{23}$$

*Proof.* Upper bounding the expression $\sum_t s_{a(t)}(B(t))$ follows by the same steps in Chu et al. [2011, Lemma 3] for the matrix $\mathcal{B}(B(t))$ (we loose a constant factor in the process). The reason is that the term $\sum s_{B(t),a(t)}$ can be written in terms of eigenvalues of $\mathcal{B}(B(t))$ matrices. More precisely, from Lemma 2 in Chu et al. (2011) we can arrange eigenvalues of $\mathcal{B}(t)$ to obtain the following bound

$$s_{a(t)}(B(t))^2 \leq 10 \sum_j \frac{\lambda_{t+1,j} - \lambda_{t,j}}{\lambda_{t,j}}.$$

Note that the above upper bound is independent of our batch algorithm. Then for $\psi = |\Psi_{T+1}|$ (in Chu et al. [2011, Lemma 3]) we have

$$\sum_{t \in \Psi_{T+1}} s_{a(t)}(B(t)) = \sum_{t \in \Psi_{T+1}} \sqrt{10 \sum_j (\frac{\lambda_{t+1,j}}{\lambda_{t,j}} - 1)},$$

for each matrix $\mathcal{B}(B(t))$ in $\Psi_{T+1}$. The function $f$ can be defined similar to Chu et al. [2011, Lemma 3] for $\Psi_{T+1}$. As in Lemma 3, the ratio of eigenvalues remain greater than or equal 1. The following sum product can be bounded by $\psi + d$ since the norm of each $x_{ta(t)}$ is bounded by 1. For $t' = B(t)$ between $T/2$ and $T + 1$,

$$\sum_j \prod_t \frac{\lambda_{t+1,j}}{\lambda_{t,j}} \leq \sum_j \lambda_{t',j} = \sum_t \|x_{t,a(t)}\|^2 + d \leq \psi + d.$$

So, we can similarly bound

$$\sum_{t \in \Psi_{T+1}} s_{a(t)}(B(t)) \leq \psi\sqrt{10d}\sqrt{(\psi+1)^{1/\psi} - 1}.$$

Thus, by using Chu et al. [2011, Lemma 9] for $\psi$ we can obtain eq. (23). $\quad\square$

*Proof of Theorem 6.1.* We rely on the proof technique by Agrawal and Goyal [2013b, Theorem 1]. First, note that $X_t$ is bounded as

$$|X_t| \leq 1 + \frac{3}{p}g(t) + \frac{2}{pt^2}g(t) \leq \frac{6}{p}g(t).$$

Also $g(t) \leq g(T)$. Thus, by applying Azuma-Hoeffding inequality for Martingale sequences, we have

$$\Pr\left(\sum_{t=1}^{T} \mathcal{R}'(t) \leq \frac{3g(T)}{p}\sum_{t=1}^{T} s_{a(t)}(B(t)) + \frac{2g(T)}{p}\sum_{t=1}^{T} \frac{1}{t^2} + \frac{6g(T)}{p}\sqrt{2T \ln(2/\delta)}\right) \geq 1 - \frac{\delta}{2}.$$

Therefore, by invoking Lemma C.18 we know that with probability $1 - \frac{\delta}{2}$ we have

$$\sum_{t=1}^{T} \mathcal{R}'(t) = O\left(d\sqrt{T} \times (\min\{\sqrt{d}, \sqrt{\log N}\}) \times (\ln(T) + \sqrt{\ln(T)\ln(1/\delta))}\right).$$

Furthermore, Lemma C.11 implies that with probability of at least $1 - \delta/2$, the event $E^{\mu}(t)$ holds for all $t$. Thus, with probability of at least $1 - \delta$,

$$\mathcal{R}(T) = O\left(d\sqrt{T} \times (\min\{\sqrt{d}, \sqrt{\log N}\}) \times (\ln(T) + \sqrt{\ln(T)\ln(1/\delta))}\right).$$

$\square$