# OpenReview forum: "Parallelizing Thompson Sampling"
_NeurIPS.cc/2021/Conference — NeurIPS 2021 Poster_

### Official Review · Reviewer_tYDL · 2021-07-01

**Rating:** 8
**Confidence:** 4

**Summary:**

The paper "Parallelizing Thompson Sampling" presents a batch-version of the traditional sequential Thompson Sampling which uses maximum $O(\textrm{Number-of-Arms} \times \log(T))$ batches to interact with the environment.

The paper shows that this is only $O(\textrm{Number-of-Arms} \log\log T)$ worse than the minimum possible number of batches, but it achieves the same regret as traditional Thompson Sampling, which can be thought of as using $O(T)$ batches of size 1. The key idea in the algorithm is that the batch sizes are randomized. A new batch is started as soon as we approximately double the number of queried examples since the last batch.

This paper is a natural expansion of the recent line of work on Batch-UCB to Thompson Sampling and I believe it will be interesting to audience of NEURIPS.


**Ethical Concerns:**

No ethical concerns.

**Limitations And Societal Impact:**

Yes. The discussion was adequate.

**Main Review:**

**Originality**: Batch version of classic bandit problems have emerged fairly recently, with most of the work done on UCB style approaches. This paper presents novel Thompson Sampling algorithms for batch multi-armed bandit, and batch linear contextual bandit.

**Quality and Clarity**

I found this paper to be well written and self-contained. Specially the Related Work section which nicely laid out the existing results in this area. I spent quite some time with Appendix A.2 which proves the main theorem of the paper and I think the proof is correct and fairly well - presented. I really liked the clear definitions in Appendix A.1 as well. That said here are some corrections and small suggestions for improvement.

1. The statement of Thm 4.3 can be made to match the traditional $\displaystyle\sum\frac{1}{\Delta}$ regret bound by using the pinsker inequality.

Let the equation between eq(6) and eq(7) be (6.5)

2. Please use $\mathbb{I}$ consistently in equation (6.5c) and (6.5d)

3. The inequality sign should actually be equal sign in (6.5d)

4. Eq(7) should use $p_{a, \tau_k + 1}$ instead of $p_{a, \tau_k} + 1$ in denominator. Same for Lemma A.11

5. The summation in the first inequality in Eq(7) should be from $0$ to $L_a(T)$ not from $0$ to $T-1$


**Significance** Although I doubt that the paper will have much of a practical impact on practitioners, but I think it will be well-received by theory researchers. I will like to propose some additional experiments to clarify the importance of the empirical results.

1. Figure 2(b) compares the performance of the proposed algorithm to a "naive" batch-TS algorithm with a fixed batch-size. This comparison is helpful, but I will also a comparison to a scheme with exponentially increasing batch-sizes on a fixed schedule. Something like the epoch greedy schedule that partitions the time-steps into exponentially bigger intervals like {1, 2, 4, 8, 16, 32, 64, 128, ...}. This will use only $\log(T)$ batches and spend more time exploring in the beginning and should have much better regret than the naive baseline presented in the paper.

2. I will also like to see the empirical histogram of batch-sizes and the number of batches across various runs of this algorithm. Having tight control -- concentration -- over the batch sizes and number of batches may be necessary for practical deployment.

**Time Spent Reviewing:**

3

---

> ### Author Response · Authors · 2021-08-07
> **Response to Reviewer tYDL**
>
> Thank you very much for your encouraging remarks and detailed comments. We tried our best to write a clear paper with sufficient references to all the amazing previous work.
>
> > The statement of Thm 4.3 can be made to match the traditional $\sum\frac{1}{\Delta}$regret bound by using the pinsker inequality.
>
> Great suggestion.
>
> > Please use I consistently in equation (6.5c) and (6.5d)
>
> Thanks. We will fix the typo.
>
> > The inequality sign should actually be equal sign in (6.5d)
>
> You are right. We will fix it.
>
> > Eq(7) should use  $p_{a,\tau_k+1}$ instead of  $p_{a,\tau_k}+1$  in denominator. Same for Lemma A.11
>
> You are right. We will fix the typo.
>
> > The summation in the first inequality in Eq(7) should be from  $0$ to  $L_a(T)$ not from $0$ to $T-1$
>
>  You are right again. We will fix it.
>
> > Figure 2(b) compares the performance of the proposed algorithm to a "naive" batch-TS algorithm with a fixed batch-size. This comparison is helpful, but I will also a comparison to a scheme with exponentially increasing batch-sizes on a fixed schedule. Something like the epoch greedy schedule that partitions the time-steps into exponentially bigger intervals like $\{1, 2, 4, 8, 16, 32, 64, 128, ...\}$. This will use only $\log⁡(T)$ batches and spend more time exploring in the beginning and should have much better regret than the naive baseline presented in the paper.
>
>
> This is a great suggestion. We will try to add this baseline to the final version.
>
>
> > I will also like to see the empirical histogram of batch-sizes and the number of batches across various runs of this algorithm. Having tight control -- concentration -- over the batch sizes and number of batches may be necessary for practical deployment.
>
> Great suggestion. We will try to add a histogram to the final version.

---

### Official Review · Reviewer_E33d · 2021-07-13

**Rating:** 4
**Confidence:** 5

**Summary:**

This paper introduces a batch framework of Thompson Sampling for the stochastic multi-armed bandit and linear contextual bandit, which achieves the same regret bound of a fully sequential one while carrying out only O(log T) batch queries.

**Limitations And Societal Impact:**

Yes

**Main Review:**

The idea of the proposed batch framework has already been developed and analyzed for finite horizon MDP, named by stage-based update framework in Zhang et al., 2020. Same as Algorithm 1, the sample rule of stage-based update remains unchanged until some state action pair is visited (1+1/H)^k times, which is shown can achieve low switching cost.
The finite horizon MDP class contains the stochastic multi-armed bandit as a special case by taking the state space size S = 1 and horizon H = 1. Although Zhang et al. consider the UCB update rule, different from Thompson Sampling herein, this makes little difference about the nature of batch policies can achieve similar regret bounds but with fewer number of interactions with the environment. Hence, I think the novelty of this paper is not enough. In addition, given the results about UCB update for finite horizon MDP (including bandit), the importance of this paper is also insufficient.
I would raise my score if the significance and difficulties of this extension from UCB update for finite horizon MDP to Thompson Sampling for bandit can be explained well.

Zhang, Z., Zhou, Y., and Ji, X. (2020). Almost optimal model-free reinforcement learning via reference-advantage decomposition. Advances in Neural Information Processing Systems, 33.

**Time Spent Reviewing:**

2

---

> ### Author Response · Authors · 2021-08-07
> **Response to Reviewer E33d**
>
> We thank the reviewer for bringing up an important point of confusion that should be further explained in the final version of our paper.
>
> There are indeed two popular notions of limited adaptivity models: rare policy switch model} and batch learning model. In the rare policy switch model, the agent continuously observes the rewards of her actions, based on which she can decide when to switch her policy. The agent is only restricted by the number of policy switches she carries out. The switching constrained model has been extensively studied in the bandit (see for example “Online learning with switching costs and other adaptive adversaries”, NIPS 2013) and more recently in the MDP literature. We will cite Zhang et al 2020 and the related papers in the final version. In contrast, in the batch setting, the agent does not observe the rewards of her actions until at the end of a batch and then observes all of them simultaneously. Moreover, the agent may not be constrained by the number of switches and in fact can run multiple different policies within a batch for better exploration. The batch learning model has recently received a lot of attention (see for example “Batched Multi-armed Bandits Problem”, NeurIPS 2019).  These two settings capture different aspects of adaptivity and  thus results from one cannot be readily applied to another.
>
> To make the distinction more clear in the bandit setting, the switching constrained setting usually refers to the case where the policy selects a single action for a period of time (while observing instantaneous rewards for each action) and then based on the observations decides to switch to a new action. It is long known that $\log T$ switches suffice to achieve an asymptotic optimal regret in this setting (see for example “Prediction by random-walk perturbation”, ALT2013).  In contrast, in the batch model, the agent may select different actions within a batch (like our batch TS policy or the existing batch arm elimination policies) but receive the rewards only at the end of the batch. We show that indeed $\log T$ batches of observations suffice to achieve a comparable regret to the fully sequential TS setting.
>
> We should also highlight the fact that the proof techniques underlying batch arm elimination and our proposed batch Thompson sampling are quite different. For instance, the recently proposed algorithm in “Regret Bounds for Batched Bandits” first pulls each active arm an equal number of times within each batch. Once the rewards are observed the sub-optimal arms are estimated (using standard concentration bounds) and then eliminated. Our batch Thomson sampling method works quite differently and the analysis is much more involved.
>
> We very much hope that given the above clarifications, the reviewer may revise their score.

---

> > ### Comment · Reviewer_E33d · 2021-08-17
> > **Misunderstanding of Zhang et al 2020 (arXiv:2004.10019)**
> >
> > You said "based on the observations decides to switch to a new action". This is a clear misunderstanding. In fact, Algorithm 1 in Zhang et al 2020 keeps the values of Q-function until some state action pair is visited (1+1/H)^k times (see Eq. (9)), which means that the policy will only change at the end of stages. Particularly, the policy switch will not change within some stage based on instantaneous observed rewards.
> >
> > In addition, you said "In the rare policy switch model, the agent continuously observes the rewards of her actions". This seems to exaggerate their difference. One can check that only (4)(5)(6) in Algorithm 1 in Zhang et al 2020 is updated continuously. Actually, this is designed only for ease of presentation. The updates (4)(5)(6) can also be done at the end of the stage when observing all of values simultaneously (similar to the rewards for bandits).
> >
> > Given the above facts, I will keep my score.

---

> > > ### Author Response · Authors · 2021-08-17
> > > **Policy Switch vs Batch Setting**
> > >
> > > We thank the reviewer for their comments about Zhang et al 2020.
> > > We firmly stand by the differences between the two settings as they are pretty established scenarios in the literature. The reviewer can simply check the following recent papers:
> > >
> > > "Sequential batch learning in finite-action linear contextual bandits", arXiv preprint arXiv:2004.06321
> > > "Linear bandits with limited adaptivity and learning distributional optimal design" arXiv:2007.01980
> > >
> > > As it is currently presented in Zhang et al 2020, the algorithm continuously observe the rewards. However, we agree that it may not need to do so and can make the switch based on the number of visits to a certain action-state. What we fully fail to understand is how this is related to batch Thompson Sampling (the focus of this paper). We have already cited previous work, based on UCB and arm elimination, that achieve low regret with $\log T$ number of batches (Zhang et 2020 belong to the same category and we will cite them in the final version). The focus of our paper is to design a batch policy for Thompson Sampling. We highly appreciate it if the reviewer can explain how the paper by Zhang et al 2020 can lead to a batch  Thompson Sampling algorithm. The reviewer should certainly know that even in the linear contextual setting, designing a Thompson Sampling algorithm that achieves the tight $O(d\sqrt{T})$ regret bound is not settled. Even more related to the paper by Zhang et al 2020, currently, there is no Thompson Sampling algorithm beyond tabular RL setting.

---

> > > > ### Comment · Reviewer_E33d · 2021-08-18
> > > > **Policy Switch in Zhang et al 2020 vs Batch Setting**
> > > >
> > > > My main concerns are still not changed.
> > > >
> > > > First, compared with Zhang et al 2020, the result here has no improvement in terms of regret bounds and number of interactions with the environment for stochastic bandits. Hence, the importance of this paper is insufficient.
> > > >
> > > > Second, applying the stage-based update, the idea introduced in Zhang et al 2020, to TS gives batch TS proposed here directly. This shows that the novelty is not enough (at least in the level of algorithm design).
> > > >
> > > > Given the above two points, I think the current version, which needs a major revision, is not ready to be accepted by NeurIPS.
> > > >
> > > > In addition, I also have some minor concern about the difference of analysis. I'm not sure about this point, since this part is in Appendix, which I haven't checked carefully.

---

> > > > > ### Author Response · Authors · 2021-08-18
> > > > > **Thompson vs UCB**
> > > > >
> > > > > We thank the reviewer for their prompt response.
> > > > >
> > > > > > First, compared with Zhang et al 2020, the result here has no improvement in terms of regret bounds and number of interactions with the environment for stochastic bandits. Hence, the importance of this paper is insufficient.
> > > > >
> > > > > Theorem 4.3 of our paper shows that B-TS achieves the optimal problem-dependent regret in  $O(\log T)$ number of batches. Theorem 3 (Corollary 2) in Gao et al 2019  shows that to achieve the optimal problem-dependent regret bound, any batch algorithm has to have $O(\log T/\log\log T)$ number of batches. So in terms of $T$, our batch algorithm is only $O(\log\log T)$ away from the lower bound. Indeed, none of the existing algorithms, e.g., batch arm elimination or batch UCB, achieve the lower bound of  $O(\log T/\log\log T)$ for the optimal problem-dependent regret bound. This point was mentioned in Remark 4.5 of our paper. In summary, the reviewer is asking for an improvement (beyond $O(\log\log T)$ ) that cannot be achieved based on the well established lower bounds.  In this work, we aim to design a Thompson Sampling mechanism that achieves the same trade-offs that exist for UCB-like algorithms.
> > > > >
> > > > > Another important issue is that the result of Zhang et al 2020 cannot even be applied to the linear contextual setting where the contexts are chosen adversarially.  In section 6 of our paper, we have developed a batch bandit algorithm for precisely this scenario.
> > > > >
> > > > > > Second, applying the stage-based update, the idea introduced in Zhang et al 2020, to TS gives batch TS proposed here directly. This shows that the novelty is not enough (at least in the level of algorithm design).
> > > > >
> > > > > This issue might be the main confusion of the reviewer. The result of Zhang et al 2020 leads to the following batch UCB algorithm in the bandit setting.  Consider the version of UCB that works in phases of exponentially increasing length of $1, 2, 4,  \dots$. In each phase, the algorithm selects the action that would have been chosen by UCB at the beginning of the phase and keeps it for the entire duration of the phase. It is very well known that such an algorithm achieves the same regret bound of UCB with $O(\log T)$ batches (please see Exercise 7.4, 7.5 in the book "Bandit Algorithms" https://tor-lattimore.com/downloads/book/book.pdf).
> > > > >
> > > > > Indeed, for Thompson Sampling if you keep the same action for the entire duration of the batch (as suggested by the reviewer), (empirically) it results in much higher ( and potentially linear) regret. This is precisely the reason we develop a quite different mechanism in our paper. It does not select the same action during a batch. Instead, it samples different arms based on the posterior distributions calculated from the previous batches and keeps count of how many times an arm is selected in order to terminate the batch (this is usually referred to as static versus dynamic batch design in the literature). This is indeed crucial to achieve optimal regret with $O(\log T)$ batches and it is specifically designed for the Thompson Sampling. We can add the empirical result in the final version and we very much hope that the reviewer appreciates the subtlety and the difference between the two methods, one that leads to a high regret (suggested by the reviewer) and ours that leads to the optimal regret.

---

### Official Review · Reviewer_2HGA · 2021-07-16

**Rating:** 6
**Confidence:** 4

**Summary:**

Paper introduces a batch Thompson Sampling framework, and studies it for stochastic multi-arm bandit and linear contextual bandit with finitely many arms. They show that it achieves the same minimax and asymptotic regrets as the standard sequential algorithms.

**Main Review:**

Two major paper strengths are that the paper is well written, and its strong theoretical and empirical results.

The paper's contribution is incremental. The proofs closely follow proofs of other papers, with minor changes to accommodate batches. The reason dynamic batching is not affecting regret guarantees seem to stem from the doubling trick (see https://arxiv.org/pdf/1603.06560.pdf).

In fact, bandits leveraging the doubling trick, like for transductive reasoning, also achieve similar batching. Albeit in their use-case, batching is a by-product, and not necessarily the objective. See:
- http://proceedings.mlr.press/v139/jun21a/jun21a.pdf (ICML'21)
- https://arxiv.org/abs/1906.08399 (NeurIPS'19)

On the practical side, it is unclear to me what actual use-cases would want such a dynamic batching for the sake of batching. My experience has been that people batch because they are forced to, due to delays and system issues (like production system updates once a day). Possible a reflection of that, and a discussion of the reasons the doubling trick achieves the desired result, could enrich the paper.

Finally, the parallelizing claim is a little misleading. although I get the author's points, still typically one thinks of parallelizing as parallelizing batch trainings. The authors refer to basic sequential batching, on one machine, as parallelizing.

Overall, the paper still has merits, in the sense that it states batch-related theorems that others could leverage, cite or build on, all in one paper. It is likely though that similar theorems existed implicitly in other papers, like the ones I reference above.

Minor:
Thrm 5.1. Delta_a not previously defined in paper.
Since the appearance of Delta (gap) in a minimax theorem may sound counter-intuitive, you may want to remove it from the main theorem's formulation, and explain it in the appendix.
For instance, you can include similar language to Jin et al 2020, in their paragraph immediately below equation (5) of their paper.




**Time Spent Reviewing:**

5

---

> ### Author Response · Authors · 2021-08-07
> **Response to Reviewer 2HGA**
>
> Thank you for your detailed comments and pointers to related work.
>
> >  The paper's contribution is incremental. The proofs closely follow proofs of other papers, with minor changes to accommodate batches.
>
> We agree that the skeleton of the proofs follows the previous work, albeit with careful adjustments. We believe that the simplicity of our algorithm and how it naturally builds upon the previous work is an advantage.  We have clearly pointed out all the nuances and differences in our proofs.
>
> > The reason dynamic batching is not affecting regret guarantees seem to stem from the doubling trick (see https://arxiv.org/pdf/1603.06560.pdf).
>
> We would also like to thank the reviewer for pointing out the above paper (we will cite it in the final version). Even though the topic is related, the setup differs. In particular, the above paper considers the pure-exploration bandit. It is a fully sequential setting (rewards are immediately observed) with a different goal, namely, identifying the best arm with the tightest confidence bound. In our paper, we aim to parallelize the Thompson Sampling procedure where the rewards are not immediately revealed (they are revealed simultaneously when the batch terminates) and the goal is to minimize the accumulated regret. We believe that it is an interesting future direction to develop batch Thompson sampling for the pure-exploration setting.
>
> > In fact, bandits leveraging the doubling trick, like for transductive reasoning, also achieve similar batching. Albeit in their use-case, batching is a by-product, and not necessarily the objective. See:
> http://proceedings.mlr.press/v139/jun21a/jun21a.pdf (ICML'21)
> https://arxiv.org/abs/1906.08399 (NeurIPS'19)
>
> We thank the reviewer for bringing these papers to our attention and we will cite them in the final version. Again, the bandit setting considered in these papers is quite different. In both cases, the pure-exploration with the goal of finding the best arm is addressed.
>
> Doubling vs Halving: Technically, the doubling mechanism in our paper and the halving trick in pure-exploration are somewhat different. The doubling mechanism is used to increase the duration of each batch as we obtain more information as the posterior distributions of arms concentrate more around the mean rewards. Note that no arm is being eliminated. In contrast, the halving mechanism aims to eliminate the arms with suboptimal rewards in a greedy fashion in order to find the near-optimal arms.  However, we agree that the techniques are related.
>
> > On the practical side, it is unclear to me what actual use-cases would want such a dynamic batching for the sake of batching.
>
> A simple example is A/B testing where the mechanism is usually under the designer's control. In this application, there are two clear procedures: non-adaptive and fully adaptive policies. For the sake of simplicity, let’s assume that each user takes 1 second to like/dislike. In a non-sequential setting, all the queries are asked in parallel among T users where half the population are given option A and the other half option B. Since everything is run in parallel it only takes one second to finish the whole experiment. The main issue is that half the population receive the suboptimal choice. In a fully sequential setting, users sequentially receive the option A or B, based on the previous responses. This experiment takes T seconds to finish. However, the benefit is that many users receive the better option. What we propose in this paper is a solution where the whole experiment takes $O(\log T)$ seconds while many users benefit from the guarantee of a fully adaptive policy.
>
> >  The parallelizing claim is a little misleading. although I get the author's points, still typically one thinks of parallelizing as parallelizing batch trainings. The authors refer to basic sequential batching, on one machine, as parallelizing.
>
> We are following the terminology of the prior work such as “Parallelizing Exploration-Exploitation Tradeoffs in Gaussian Process Bandit Optimization”. Parallelization in this context means that all the arms within a batch can be carried out simultaneously and in parallel.
>
> > Overall, the paper still has merits, in the sense that it states batch-related theorems that others could leverage, cite or build on, all in one paper. It is likely though that similar theorems existed implicitly in other papers, like the ones I reference above.
>
> Thank you for your encouraging comments. As we stated before, the pure-exploration setting is quite different from the batch setting. To the best of our knowledge, we are the first to propose a batch Thompson sampling policy with strong theoretical guarantees.
>
> > Thrm 5.1. Delta_a not previously defined in paper.
>
> $\Delta_a$ was defined in Section 3, line 136. Thanks for the suggestion. We will add an explanation in the final version.

---

### Official Review · Reviewer_Qcd6 · 2021-07-17

**Rating:** 7
**Confidence:** 2

**Summary:**

This paper investigates dynamic-batch Thompson sampling. The proposed methods enjoys enjoy theoretically preferable regret bounds. Particular, in the stochastic multi-armed bandit setting,

* B-TS achieves the problem-dependent asymptotic optimal regret;
* B-MOTS achieves the optimal minimax problem-independent regret bound;
* B-MOTS-J, designed for Gaussian rewards, achieves both minimax and asymptotic optimality.

Moreover, a method named B-TS-C is also proposed applicable in the linear contextual bandit setting.

Finally, the proposed methods are compared with their fully sequential counterparts via experiements on synthetic and small real data.

**Ethics Review Area:**

["I don’t know"]

**Main Review:**

**Originality**

This extends existing batch UCB algorithms to the case of Thompson Sampling for the stochastic multi-armed bandit as well as the linear contextual bandit problems.

**Clarity**

This paper is clearly presented and readable. Although I am not very familiar with this research area, I could keep up with the storyline without much effort.

**Significance**

As pointed out in the Summary section, in the stochastic multi-armed bandit setting,

* B-TS achieves the problem-dependent asymptotic optimal regret;
* B-MOTS achieves the optimal minimax problem-independent regret bound;
* B-MOTS-J, designed for Gaussian rewards, achieves both minimax and asymptotic optimality.

**Limitations**

B-TS-C seems (I am not sure and the paper does not claim explicitly) to be the first dynamic-batch Thompson sampling method for the linear contextual bandit setting. However, while most of the papes of the paper are about the stochastic multi-armed bandit, it is suggested to add more discussion about B-TS-C, such as the adventages, limitations and future work.

**Time Spent Reviewing:**

4

---

> ### Author Response · Authors · 2021-08-07
> **Response to Reviewer Qcd6**
>
> Thank you very much for your encouraging remarks.
>
> >  B-TS-C seems (I am not sure and the paper does not claim explicitly) to be the first dynamic-batch Thompson sampling method for the linear contextual bandit setting. However, while most of the papes are about the stochastic multi-armed bandit, it is suggested to add more discussion about B-TS-C, such as the advantages, limitations and future work.
>
> Indeed, to the best of our knowledge, B-TS-C is the first dynamic-batch Thompson sampling method for the linear contextual bandit setting. The multi-armed bandit sections mainly served as the foundations for developing efficient dynamic batch Thompson sampling. We agree with the reviewer that in many online learning applications, we usually encounter a contextual setting which may require further explanations. One important future direction in linear contextual setting is to reduce the dimension dependence from $d^{3/2}$ to $d$. Another important direction is to extend the result to the case where we have infinitely many arms (similar to batch arm elimination).

---

### Decision · Program_Chairs · 2021-09-28

**Decision:**

Accept (Poster)

**Comment:**

This paper proposes several _batched_ Thompson sampling (TS) algorithms for stochastic and linear contextual bandit problems. In this setting, the bandit may take multiple actions before updating. For instance, if the bandit were optimizing a recommender system, it could serve recommendations to multiple users before collecting feedback and updating. Importantly, the bandit gets to decide how many actions it takes in each batch. Its goal is to minimize its regret _as well as the number of batches_. The proposed algorithms all achieve the optimal $\tilde{O}(\sqrt{T})$ (sublinear) regret using only $\tilde{O}(\log T)$ batches. Experiments show that the batched TS algorithms perform as well as fully sequential TS and better than UCB.

The paper is clear, well written and easy to follow. The theoretical results are strong in the sense that they achieve optimal rates. The empirical results are thorough and convincing. It's a nice paper that will be of interest to the NeurIPS community.

The reviews raise the following concerns.

(1) The work is closely related to recent work by Zhang et al. (NeurIPS, 2020) which proposed a batch UCB-style algorithm for tabular MDPs. The asymptotic rates for batched TS are no better than those for batched UCB. For this reason, one reviewer argued that the current work is incremental and not worth accepting. During discussion, the following points (paraphrased) were made by another reviewer:

* Zhang et al.'s work only applies to tabular MDPs with discrete states; hence, it doesn't apply to contextual bandits.
* Since Zhang et al.'s work studies MDPs, it can be thought of operating in a stochastic reward setting instead of adversarial. The current paper accommodates an adversarial setting.
* The fact that the proposed TS algorithms cannot guarantee better asymptotic rates is not a reason to reject. Aggarwal et al.'s seminal papers analyzing TS did not show better rates than UCB; just that TS performed close to optimal, off by a log factor. TS and UCB operate differently and as such their analyses are different, and historically advancements in either algorithms have merited publication.

Based on these points, I do not think Zhang et al.'s work subsumes the current work, and I do not see this issue as grounds for rejection.

(2) The analysis techniques are well known -- in particular, the "doubling trick" has been used in the past (see Review 2HGA), albeit not with batching in mind -- which detracts from the paper's contributions. Nonetheless, the combination and application of these techniques is compelling. And, as Reviewer 2HGA points out, it's nice to have all of these batching theorems compiled in a single paper, for reference by others.

(3) The practicality of the proposed algorithm is questionable, due to the assumption that the algorithm gets to decide how much to batch. In a typical bandit application (e.g., a recommender system or ad placement system), batching is used out of necessity, due to external factors like latency or limited computing resources.

I am recommending Weak Accept, because it is a nice paper, and I believe this paper will be of value to the NeurIPS community (perhaps mainly theoreticians), but the practicality of the proposed algorithms is questionable.

**Consistency Experiment:**

NeurIPS has a long history of experimentation. In 2014, NeurIPS ran an experiment in which 10% of submissions were reviewed by two independent committees to quantify the randomness in the review process. This year, we repeated a variant of this experiment to see how the quality of the review process has changed over time.  This paper was part of the experiment and was therefore assigned to two committees (consisting of reviewers, an Area Chair, and a Senior Area Chair) that reached independent decisions.  If both committees made the same recommendation, this recommendation was followed. If a single committee recommended acceptance, the paper was accepted (with the exception of a few cases in which the other committee identified what we considered a fatal flaw, e.g., an error in a key result).

This copy’s committee reached the following decision: **Accept (Poster)**

The other committee assigned to the paper recommended **Reject**.  You can find the other set of reviews, along with any follow up discussion with the authors here:
https://openreview.net/forum?id=rdMQrE-loT5